# Overall water splitting by graphdiyne-exfoliated and -sandwiched layered double-hydroxide nanosheet arrays

Lan Hui[1,2], Yurui Xue [1,3], Bolong Huang [4], Huidi Yu[1], Chao Zhang[1]
Danyan Zhang[1], Dianzeng Jia[2], Yingjie Zhao[1,3], Yongjun Li [1], Huibiao Liu[1] & Yuliang Li [1,3,5]

It is of great urgency to develop efficient, cost-effective, stable and industrially applicable electrocatalysts for renewable energy systems. But there are still few candidate materials. Here we show a bifunctional electrocatalyst, comprising graphdiyne-exfoliated and -sandwiched iron/cobalt layered double-hydroxide nanosheet arrays grown on nickel foam, for the oxygen and hydrogen evolution reactions. Theoretical and experimental data revealed that the charge transport kinetics of the structure were superior to iron/cobalt layered double-hydroxide, a prerequisite for improved electrocatalytic performance. The incorporation with graphdiyne increased the number of catalytically active sites and prevented corrosion, leading to greatly enhanced electrocatalytic activity and stability for oxygen evolution reaction, hydrogen evolution reaction, as well as overall water splitting. Our results suggest that the use of graphdiyne might open up new pathways for the design and fabrication of earth-abundant, efficient, functional, and smart electrode materials with practical applications.

[1] Key Laboratory of Organic Solids, Institute of Chemistry, Chinese Academy of Sciences, Beijing 100190, P. R. China. [2] Key Laboratory of Energy Materials Chemistry, Ministry of Education; Key laboratory of Advanced Functional Materials, Autonomous Region; Institute of Applied Chemistry, Xinjiang University, Urumqi 830046, P. R. China. [3] School of Polymer Science and Engineering, Qingdao University of Science and Technology, Qingdao 266042, P. R. China. [4] Department of Applied Biology and Chemical Technology, The Hong Kong Polytechnic University, Hong Kong, P. R. China. [5] University of Chinese Academy of Sciences, Beijing 100049, P. R. China. Correspondence and requests for materials should be addressed to Y.X. (email: xueyurui@iccas.ac.cn) or to D.J. (email: jdz@xju.edu.cn) or to Y.L. (email: ylli@iccas.ac.cn)

The electrochemical or photoelectrochemical splitting of water into hydrogen and oxygen is a sustainable technology for energy conversion and storage[1–4]. Among the drawbacks of applying these approaches industrially are the sluggish kinetics of the oxygen and hydrogen evolution reactions (OER/HER). Thus, there remains great demand for efficient, cost-effective, stable, and industrially applicable electrocatalysts for overall water splitting (OWS). At present, precious metals provide the state-of-the-art electrocatalysts for water splitting (e.g., $IrO_2$ and $RuO_2$ for OER; Pt for HER), but their use on large scales is hampered by their high cost and scarcity. Extensive efforts are, therefore, being devoted to the development of earth-abundant electrocatalysts, including first-row (3d) transitional metal (TM) layered double hydroxides (LDH)[5,6], oxides, nitrides[7], and sulfides[8]. Owing to their structural flexibility and chemical versatility, TM LDHs have emerged as promising alternatives to their precious metal counterparts, and multi-metal LDHs are particularly interesting because their intrinsic activities generally outperform those of their mono-metal counterparts. Nevertheless, the low conductivity and poor electroactivity and stability of bulk LDHs (b-LDHs) have severely impeded their potential applications[5]. Attempts have been made to solve these problems by exfoliating thick b-LDHs into ultrathin nanosheets, directly growing LDHs onto conductive substrates, or hybridizing LDHs with conductive carbon materials[5,9–12]. On the other hand, coating catalysts with two-dimensional (2D) carbon materials can also be an effective means of designing new electrocatalysts with improved activities and stabilities[12]. These approaches have been and are being used in the synthesis of LDH-based electrocatalysts, but with significant limitations. For example, although exfoliated LDHs (e-LDHs) possess more active sites and higher electronic conductivities than pristine LDHs[9], they must be bonded onto supports using polymer binders (e.g., Nafion/PTFE) when fabricating electrode materials, thereby resulting in significant degradation of the conductivity, activity, and structural stability of the electrode—the result of decreased ion/electron transport kinetics and decomposition of the catalysts. Directly growing LDHs onto substrates can obviate the need for binders[5,10,11], but the previously reported LDHs grown on substrates have always been relatively thick. Accordingly, we suspected that it would be beneficial to combine the above mentioned advantages to develop high-performance electrode materials. We are, however, to the best of our knowledge, unaware of any such previous attempts.

The all-carbon material graphdiyne (GDY)—a 2D one-atom-thick sp/sp² co-hybridized carbon network possessing silicon-like electronic conductivity[13,14], a natural band gap, and natural pores (2.5 Å)[15]—appears to be a material that has the ability to resolve several of the problems affecting current electronic and energy materials when applied in various fields (e.g., electrocatalysis, energy storage, optical devices, electronic devices), the result of its unique structural, physical, and chemical properties[16–27]. More importantly, GDY is the only carbon materials that can be synthesized in a controllable and facile manner on arbitrary substrates. In a typical reaction process, hexaethynylbenzene (HEB) monomers first interact with the substrate and self-assemble into a monolayer on its surface, followed by in situ polymerization to form GDY[16]. For LDHs, the large interlayer distance and high exchangeability of the interlayer anions make it easy for HEB monomers to enter into the LDH gallery. Coupling of the alkynyl units would then occur within the confined spaces, leading to the simultaneous formation of GDY layers on both sides of the LDH surfaces. Owing to the flexibility of both GDY and LDH, the stress/deformation caused by their intimate contact would further enlarge the layer spacing, leading to complete exfoliation of the b-LDHs into thicker LDH nanosheets.

This creative strategy has many significant advantages over more traditional ways of designing and fabricating efficient electrocatalysts. First, the intrinsically high electrical conductivity of GDY can effectively facilitate charge transport and improve electrocatalytic kinetics and performances. Another attractive property of GDY is that its high content of triple bonds makes it rich in charged carbon atoms, which can provide more active sites and, thus, better catalytic activity. Furthermore, GDY appears to be one of the most stable diacetylenic carbon materials; thus, it can effectively protect electrocatalysts from corrosion and greatly improve their long-term stability[28,29]. We suspected that our proposed in situ growth strategy would provide direct and seamless contacts between GDY and LDH, without the need for any polymer binders, thereby decreasing the resistance, improving the charge transfer behavior, and facilitating catalysis. Furthermore, the naturally porous structure of GDY and the large open spaces in the catalyst would be beneficial to efficient mass transport, gas evolution, and the efficient use of active sites, thereby greatly improving the system's catalytic performance.

Herein, we describe the in situ exfoliation and modification of bulk iron–cobalt LDH nanosheets through a GDY-induced intercalation/exfoliation/decoration strategy (e-ICLDH@GDY/NF). This approach allows us not only to exfoliate thick bulk-LDHs into ultrathin e-LDHs in situ but also simultaneously form GDY sandwiched structures. Compared with catalytically inert bulk-LDHs, the e-ICLDH@GDY/NF structure displays greatly enhanced electrocatalytic activities and stabilities for the OER and HER in 1.0 M KOH. Furthermore, this structure requires only low cell voltages of 1.43, 1.46, and 1.49 V to provide current densities of 10, 100, and 1000 mA cm$^{-2}$, respectively, for OWS.

## Results

**Synthesis and structural characterization**. Figure 1 provides a schematic representation of the synthesis of the electrocatalyst (please see Methods section for details). The self-supported iron–cobalt LDH (ICLDH) nanosheet array was first grown on nickel foam (NF) simply through hydrothermal treatment, followed by a cross-coupling reaction[30,31]. In general, the anions located in the interlayer regions can be easily replaced. And the large interlayer distance in LDH make it easy for the entrance of HEB monomers into the LDHs gallery. The acetylenic hydrogen of HEB could then form hydrogen bonds with the hydroxide layer, which can lead to the formation of HEB film on the LDH surfaces. A Glaser−Hay reaction would then take place efficiently with the aid of TMEDA. The GDY films could be uniformly grown on LDH surfaces. Owing to the flexibility nature of both GDY and LDH, the stress/deformation caused by the intimate contact in-between will further enlarge the layer spacing, leading to the complete exfoliation of b-LDHs into thicker LDH nanosheets (e-LDH). The resulted product is denoted here as e-LDH@GDY/NF.

We used scanning electron microscopy (SEM) to study the morphologies of the samples. For pure CLDH, a nanowire morphology, several micrometers long, formed on the NF surface (Fig. 2a, b); pure ILDH also had a nanosheet morphology, but with an average size of approximately 0.5 μm (Fig. 2c, d). After the incorporation of Co and Fe atoms, the ICLDH structure exhibited a nanosheet morphology similar to that of ILDH; it was grown vertically and uniformly on the surface of the NF (Fig. 2e–h). These ICLDH nanosheets possessed a smooth surface, a thickness of approximately 40–80 nm, and a larger lateral size of approximately 1.5 μm (Fig. 2h). After cross-coupling, the samples maintained their nanosheet morphology (Fig. 2i–l) but became much thinner (ca. 7.0 nm) and more wrinkled than the pristine ICLDH nanosheets. This result was further verified by atomic force microscopy (AFM) images (Fig. 2m, o) and corresponding height profiles (Fig. 2n, p). In order to confirm whether the morphologies of the LDHs in pyridine or TMEDA can be damaged, pristine LDHs were put into the reactor containing pyridine or TMEDA only (without HEB). Other experimental conditions were the same as that used

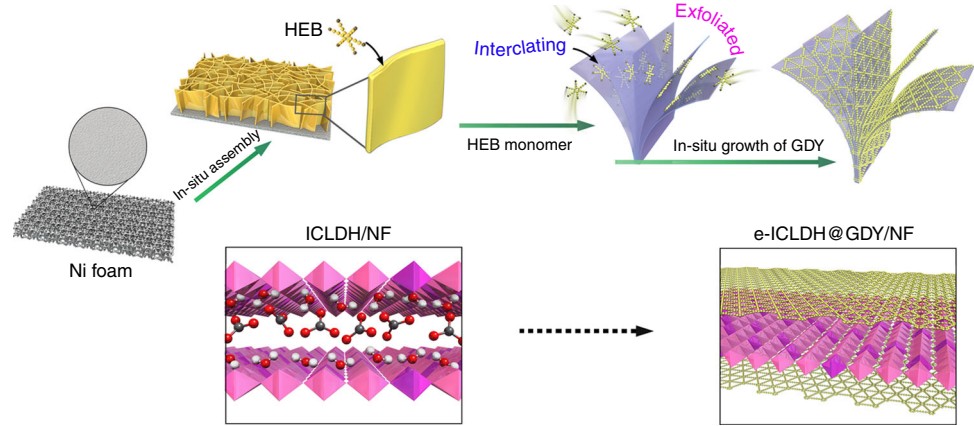

**Fig. 1** Schematic representation of the synthetic strategy for the preparation of e-ICLDH@GDY/NF structures

**Fig. 2** SEM images of LDH/NF samples. **a** Low-magnification and **b** high-magnification SEM images of CLDH/NF. Scale bars: **a** 2 μm; **b** 500 nm. **c** Low-magnification and **d** high-magnification SEM images of ILDH/NF (recorded at different areas). Scale bars: **c** 1 μm; **b** 200 nm. **e–l** SEM images of **e–h** pristine ICLDH/NF and **i–l** e-LDH@GDY/NF (recorded at different areas). Scale bars: **e** 2 μm; **f**, **g**, **j**, **k** 500 nm; **i** 1 μm; **h**, **l** 200 nm. AFM images and height profiles of **m**, **n** ICLDH nanosheets and **o**, **p** e-LDH@GDY nanosheet, respectively. Scale bars: **m** 1 μm; **o** 150 nm

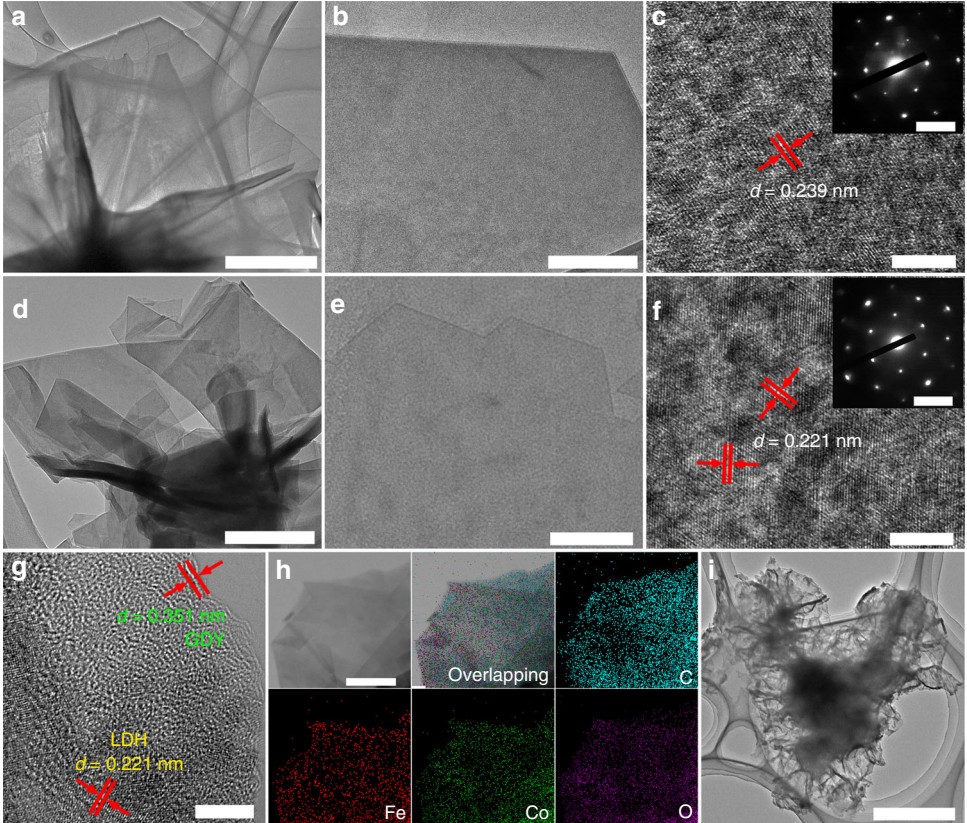

**Fig. 3** TEM characterization of electrocatalyst samples. **a** Low-magnification, **b** high-magnification, and **c** high-resolution TEM images and corresponding selected-area electron diffraction pattern (SAED, inset to **c** of pure ICLDH nanosheets. Scale bars: **a** 1 μm; **b** 300 nm; **c** 5 nm; inset of **c**, 5 1/nm. **d** Low-magnification, **e** high-magnification, and **f**, **g** high-resolution TEM images of e-ICLDH@GDY nanosheets (inset to **f** SAED pattern of e-ICLDH@GDY). Scale bars: **d** 200 nm; **e** 100 nm; **f** and **g** 5 nm; inset of **f** 5 1/nm. **h** Typical scanning TEM and corresponding elemental mapping images of C, Fe, Co, and O atoms in the e-ICLDH@GDY nanosheets. Scale bars: **h** 100 nm. **i** TEM image of the GDY coatings after removal of the ICLDH from the sample. Scale bars: **i** 500 nm

for synthesizing e-ICLDH@GDY/NF. SEM images exhibited no any changes in morphology before and after the treatments by pyridine or TMEDA (Supplementary Figs. 1 and 2). These observations indicated the exfoliation of the bulk LDHs.

Compared with the pristine ICLDH nanosheets (Fig. 3a, b), the TEM images of the e-ICLDH@GDY nanosheets (Fig. 3d, e) exhibited weak contrast, suggesting that they were very thin. The selected-area electron diffraction (SAED) patterns of the pristine ICLDH (inset to Fig. 3c) and e-ICLDH@GDY nanosheets (inset to Fig. 3f) both featured hexagonally arranged spots, revealing that the single-crystalline nature of the LDHs remained after their coating with the GDY layer. High-resolution TEM (HRTEM) images revealed that the lattice spacing of the pristine LDH decreased from 0.239 nm (Fig. 3c) to 0.221 nm (Fig. 3f) after growth of the GDY layers, suggesting strong interactions between the GDY and the ICLDH layers in the e-ICLDH@GDY. A fringe distance of 0.351 nm and a lattice spacing of 0.221 nm, corresponding to the GDY and ICLDH layers, respectively, are also be clearly evident in Fig. 3g. Supplementary Fig. 3 shows the HRTEM image of the lateral standing e-ICLDH@GDY nanosheet. The thickness of GDY and LDH are determined to be about 1.2 nm and 5.1-6.5 nm, respectively, which is in accordance with the SEM (Fig. 2i) and AFM imaging (Fig. 2o, p) results. Furthermore, scanning TEM (STEM) and energy dispersive X-ray spectroscopy (EDS) mapping images demonstrated the homogeneous distribution of Fe, Co, C, and O elements within the whole nanosheet structure (Fig. 3h and Supplementary Fig. 4). The EDS pattern (Supplementary Fig. 5)

revealed that the nanosheets contained only the elements Fe, Co, C, and O. Moreover, after treatment with concentrated nitric acid, the GDY coatings delaminated from the pristine ICLDHs and retained the basic nanosheet morphology of the LDH nanosheet array (Fig. 3i, Supplementary Fig. 6). These observations confirmed the successful growth of GDY layers on the LDH and the intimate contact between the layers of single-crystallized LDHs and GDY.

We employed Fourier transform infrared (FTIR) spectroscopy to characterize our electrocatalysts. For the e-ICLDH@GDY/NF and pristine ICLDH/NF samples, we attribute the broad intense bands between 3100 and 3600 cm$^{-1}$ to stretching of their OH groups (Fig. 4a–c). As revealed in Fig. 4b, the band at 1503 cm$^{-1}$, which generally corresponds to the stretching mode of interlayer carbonate species, is present only in the spectrum of pristine ICLHD/NF[32]. In addition, in the spectrum of e-ICLDH@GDY/NF, the bands at 1450 and 1587 cm$^{-1}$ represent skeletal vibrations of the aromatic ring, while the band at 2103 cm$^{-1}$ corresponds to the stretching of C≡C bonds (Fig. 4c)[16]. We used X-ray photoelectron spectroscopy (XPS) to obtain information regarding the elemental compositions and chemical states of our as-synthesized samples. The XPS survey spectrum of the e-ICLDH@GDY nanosheets revealed that only the elements Fe, Co, C, and O were present in the nanosheets (Fig. 4d). Compared with the XPS survey spectrum of pristine ICLDH, the absence of the peak in the range 288.0–290.2 eV implied the thorough removal of $CO_3^{2-}$ ions from the e-ICLDH structure after incorporation with GDY (Fig. 4e). Figure 4f presents the O 1 s

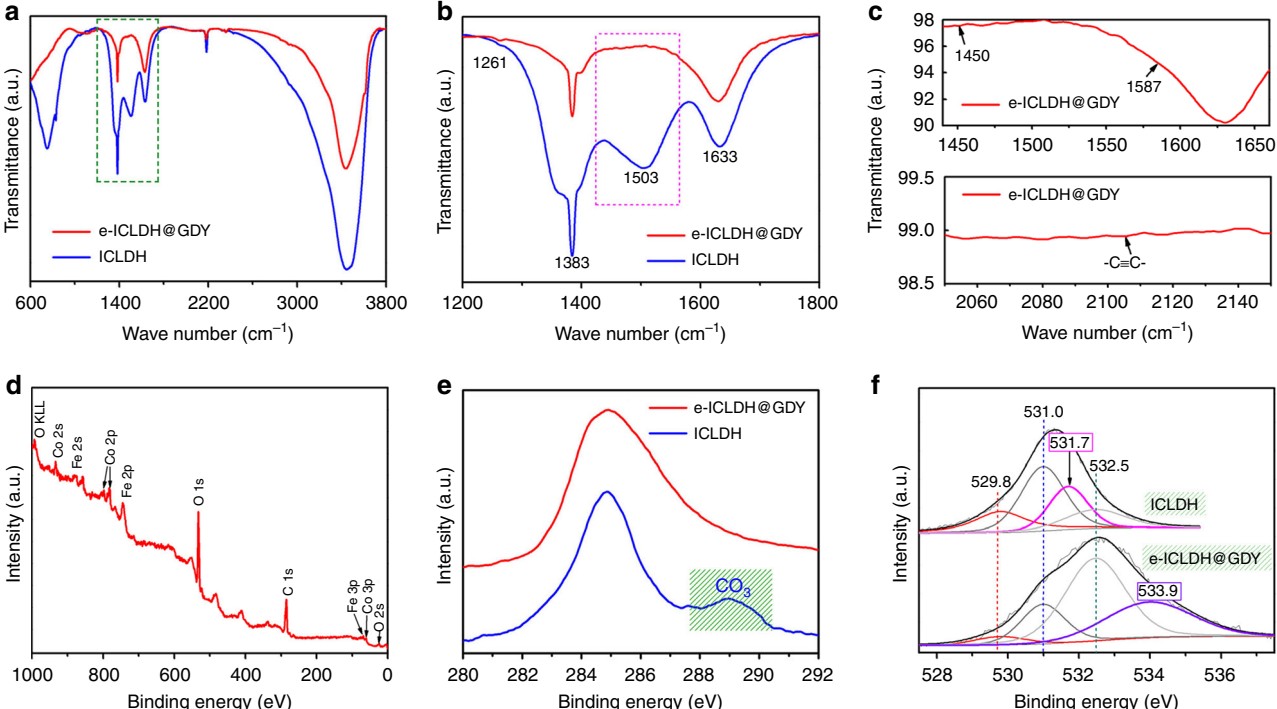

**Fig. 4** Structural characterization of electrocatalysts. **a** FTIR spectra of the e-ICLDH@GDY/NF (red line) and pristine ICLDH/NF (blue line) structures. **b** Enlarged image of the selected area of the FTIR spectrum in **a**. **c** Selective enlargement of the FTIR spectra of e-ICLDH@GDY/NF. **d** XPS survey spectrum of e-ICLDH@GDY/NF. **e, f** Core-level XPS spectra of the **e** C 1s and **f** O 1s binding energies of the e-ICLDH@GDY/NF and pristine ICLDH/NF electrocatalysts

XPS spectra of the pristine ICLDH and e-ICLDH@GDY samples. For the pristine ICLDH, we deconvoluted the O 1s spectrum into four sub-peaks at 529.8, 531.0, 531.7, and 532.5 eV, which we ascribe to lattice oxygen ($O^{2-}$), metal–oxygen bonding (M–O–M), adsorbed oxygen in the form of $CO_3^{2-}$, and OH groups, respectively[33]. In the spectrum of e-ICLDH@GDY, four peaks were present, located at 529.8, 531.0, 532.5, and 533.9 eV (Fig. 4f). We infer that the peak corresponding to $CO_3^{2-}$ ions (533.9 eV) disappeared, while a peak for C–O bonds (533.9 eV) appeared. The C–O species originated from in situ growth of GDY on the LDH surface. These spectra confirmed the removal of the interlayer anions ($CO_3^{2-}$) from the LDH and the formation of the e-ICLDH@GDY structure.

**Theoretical studies**. We performed theoretical calculations of the impact of GDY on the electronic structure of the electrocatalyst. The configurations of GDY (Fig. 5a), ICLDH (Fig. 5b, Supplementary Fig. 7 and Supplementary Methods) and ICLDH@GDY (Fig. 5c) were firstly optimized. ICLDH@GDY featured smaller lattice parameters ($a = 9.34$ Å; $b = 9.36$ Å) than did the pristine GDY ($a = b = 9.46$ Å), implying that the incorporation of LDH and GDY led to shrinkage of the GDY structure—possibly a positive influence on the catalytic activity. Furthermore, our calculations indicated that 0.80 |e| was transferred in total from ICLDH to the GDY layer, indicative of strong electronic interactions between the GDY and the LDHs (Fig. 5d). This theoretical prediction was confirmed in XPS (Fig. 5e, f) and Raman (Fig. 5g) spectra. The binding energies of both Fe $2p_{3/2}$ (Fig. 5e) and Co $2p_{3/2}$ (Fig. 5f) in the pristine ICLDH were increased by 2.2 eV (from 712.2 to 714.4 eV) and 3.0 eV (from 780.6 to 783.6 eV) after in situ growth of GDY, indicating the presence of strong electronic interactions between GDY and the metallic species in the LDHs. In the Raman spectrum of e-ICLDH@GDY (Fig. 5h), the peaks corresponding to the metal–OH stretching vibration modes

(at 448.4 and 528.5 $cm^{-1}$), the D band (1398.4 $cm^{-1}$), the G band (1576.9 $cm^{-1}$), and the vibrations of conjugated –C≡C– units (1920.6 and 2176.4 $cm^{-1}$) were shifted significantly when compared with the spectra of pristine ICLDH and GDY. We expected such associated electronic interactions between GDY and LDH to be very beneficial for the electrocatalytic performance. For the OER in alkaline electrolyte (Supplementary Fig. 8), the *O-to-*OOH conversion step is believed to be rate-determining[34]. Electrostatic interactions at the electrocatalyst interface afford a driving force for the adsorption of *O, in turn accelerating the conversion of *O to *OOH. A greater density of electrons at the electrocatalyst surface should be of more help for the formation of *OOH and thus, enhance the OER activity[34–36]. As demonstrated experimentally and theoretically above (Fig. 5), the obvious electron transfer from LDH to GDY in e-ICLDH@GDY should, therefore, favor the formation of *OOH species with a smaller free energy of 0.54 eV than that of pure GDY (0.93 eV, Fig. 5h) and benefit the OER activity. In addition, it has been reported that electrons transferred from adjacent metal species (e.g., Fe[37], Ni[37], Mo[29]) to GDY can facilitate both the adsorption (the rate-determining step) and desorption of H atoms, leading to greater HER activity.

Here we further gained some insightful interpretations on the high alkaline HER performance on the HER by DFT calculations (Fig. 5, Supplementary Fig. 9). The projected partial density of states (PDOSs) show the related p- and d-band distributions for the nearest interfacing GDY and LDH layers. The overall 3d-band center of ICLDH is high next to the Fermi level ($E_F$), which is responsible for activating the initial adsorptions of O-species from alkaline condition. The partial intrinsic overlaps of p- and d-bands demonstrate a weak inter-layer bonding. This indicates a rather long-ranged interacting interfacing layer with large distance for $H_2O$ molecule easily passing through, without obviously impacting the catalyst interface (Fig. 5i). Details on the 3d-bands of Co and Fe sites from LDH interfacing layer show

   **5**

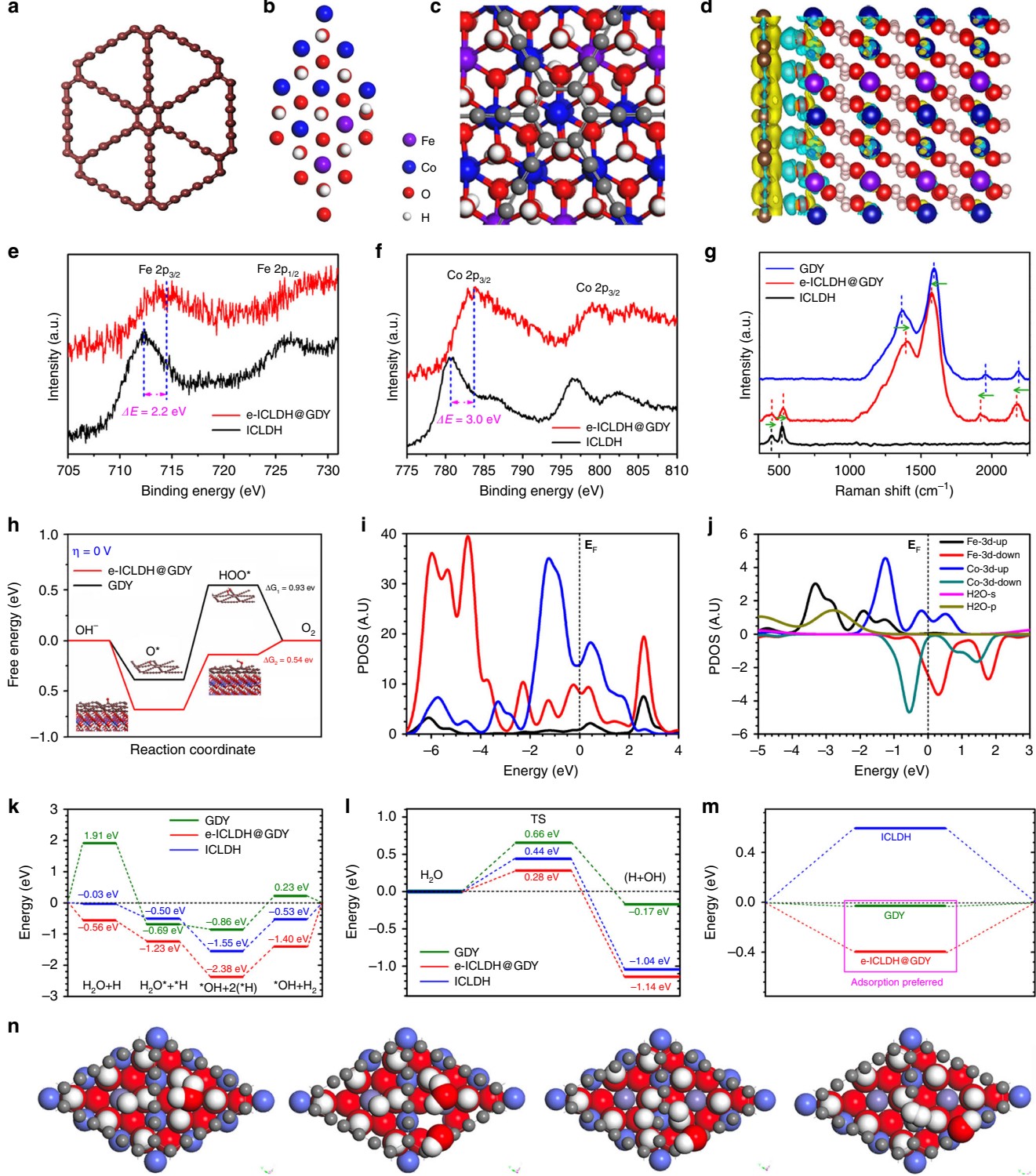

**Fig. 5** Theoretical calculations and structural analysis of the electrocatalysts. The stable configurations of **a** GDY, **b** ICLDH and **c** ICLDH@GDY. **d** Charge density difference for the stable configuration of ICLDH@GDY. **e** Fe 2p and **f** Co 2p core-level XPS spectra of the e-ICLDH@GDY/NF structure. **g** Raman spectra of GDY, ICLDH, and e-ICLDH@GDY; arrows indicate the directions of the shifts of the Raman spectral signals relative to those of e-ICLDH@GDY. **h** The free energy changes for the formation of OOH* and corresponding stable structures of GDY ($\Delta G_1$) and e-ICLDH@GDY ($\Delta G_2$). **i** PDOSs of the 3d and 2p bands of interfaced system containing GDY and ICLDH layers. **j** PDOSs of Fe-3d, Co-3d, $H_2O$-s and $H_2O$-p bands near the interface region. **k** Energetic pathway of HER under alkaline conditions for e-ICLDH@GDY, ICLDH, and GDY, respectively. **l** Comparison on the transition state barrier for $H_2O$-splitting among three systems. **m** H-chemisorption of these three systems. **n** Structural evolution path for alkaline HER within e-ICLDH@GDY interface system

the contrast in 3d-band centers, which potentially denotes a dynamically self-activated electronic activity of redox. The Co-3d bands with majority-spin is pinned at the $E_F$ and locates 1.4 eV higher than one of Fe-3d band. This trends show Fe-3d acts as electron-rich center while the Co-3d band plays as electron-depletion channel. However, their d-d overlapping is rather weak with few overlaps in PDOS. This indicates the highly effective electron-transfer occurs between the inter-layers instead of intra-layer ionic sites. The O-2p band from $H_2O$ exhibits a large overlapping across the Fe-3d band confirming the Fe-site acts as dominant role for $H_2O$ initial adsorption within interlayer regions (Fig. 5j). We move onto the energetic pathway of the alkaline HER for this interface system[38,39]. Overall, the e-ICLDH@GDY system performs the most energetic favorable path with lowest adsorption (−0.56 eV) and preferred exothermic reaction heat for $H_2$ (−1.40 eV). In the contrast, the GDY exhibits the highest barrier for initial $H_2O$ adsorption (1.92 eV) and an endothermic reaction (0.23 eV) for HER. The capabilities of water-splitting demonstrate the preference as e-ICLDH@GDY > ICLDH > GDY, respectively (Fig. 5k). We also compared the intermediating $H_2O$-spliting and the transition state barriers are 0.28 eV (e-ICLDH@GDY), 0.44 eV (ICLDH), and 0.65 eV (GDY), respectively (Fig. 5l). Meanwhile, the chemisorption energy of HER has been also illustrated. Among of them, the e-ICLDH@GDY indicates even more energetic favorable trend for HER compared with pristine-GDY, while the LDH meets the unflavored uphill for H-adsorption (Fig. 5m). From the local structures (Fig. 5n), the stable adsorbed H is bonding with C2 site at the GDY layer and the $H_2O$ location will pass through the GDY locating near Fe-site on the ICLDH surface. The splitting $H_2O$ process occurs between the C2 site and Fe-site from LDH layer. The two closely adsorbed H exhibit a favorable trend for combination in structural configuration for potentially efficient $H_2$ generation. With assistance of C2 site, the leaving group ($OH^-$) can be stably located on the surface of GDY. Accordingly, we suspected that the presence of the associated electronic interactions between GDY and ICLDH might improve the intrinsic electrocatalytic HER activity of our e-LDH@GDY/NF electrocatalyst.

**Electrocatalytic OER catalysis**. As a proof-of-concept application, we first applied the e-ICLDH@GDY/NF structure as an OER catalyst in 1.0 M aqueous KOH ($4OH^- \rightarrow O_2 + 4e^- + 2H_2O$) using a typical three-electrode system operated at 1 mV s$^{-1}$. We used cyclic voltammetry (CV) to measure the OER activities. As shown in Fig. 6a and Supplementary Fig. 10, the e-ICLDH@GDY/NF exhibited the best OER activity with the smallest overpotentials of 216, 238, 249, 275, and 278 mV at current densities ($j$) of 10, 50, 100, 500, and 1000 mA cm$^{-2}$, respectively—values that are comparable to or even better than those of previously reported LDH-related OER catalysts (Supplementary Table 1), including (Ni, Fe)OOH (289 mV at 10 mA cm$^{-2}$)[40], CoO$_x$@CN (260 mV at 10 mA cm$^{-2}$)[41], 3DGN/CoAl-NS (252 mV at 10 mA cm$^{-2}$)[11], and Co$_1$Mn$_1$CH (322 mV at 50 mA cm$^{-2}$)[10]. (Supplementary Fig. 11) The performance of e-ICLDH@GDY/NF revealed a small Tafel slope of 43.6 mV dec$^{-1}$ (Fig. 6b, Supplementary Fig. 11), much smaller than those of RuO$_2$ (76.4 mV dec$^{-1}$) and most of the reported precious and nonprecious electrocatalysts (Supplementary Table 1). From the OER results (Supplementary Fig. 10), we can see there are two major factors can make the enhanced activity. One is GDY incorporation, and another is Fe introduction into CLDH. From the polarization curves, obviously, incorporation of Fe has greater effects on the activity. As shown in Supplementary Fig. 12, the CVs for pure CLDH/NF and CLDH@GDY/NF exhibit two pronounced oxidation peaks at 1.28 and 1.36 V corresponding to the oxidation of $Co^{2+}$ to $Co^{3+}$ and the oxidation of $Co^{3+}$ to $Co^{4+}$, respectively[42-44]. For pure ILDH, the oxidation peak was

observed at round 1.41 V[45]. The CV of e-ICLDH@GDY/NF shows only one broad anodic peak at around 1.25 V and a cathodic peak at around 1.16 V (Supplementary Fig. 12d). The redox behavior of the e-ICLDH@GDY/NF sample is mainly attributed to the $Co^{2+}$/$Co^{3+}$ and $Co^{3+}$/$Co^{4+}$ redox pairs and contribution from iron[34,43,46,47]. Such high charge densities would be particularly beneficial for the improvement of the OER catalytic activity[34,43]. Theoretical studies showed the effects of Fe incorporation (Supplementary Fig. 13 and Supplementary Methods). After the configuration optimization, the calculated binding energies ($E_b$) are −2.01 eV and −2.26 eV for CLDH@GDY and ICLDH@GDY, respectively, which shows that the incorporation of Fe atoms contribute to the binding between GDY and LDH. Besides, we have also examined the charge transfer between the LDH and GDY. Compared with pure CLDH (0.70 |e|), a more electron in total (0.80 |e|) was transferred from ICLDH to GDY, indicating that the incorporation of Fe can further enhance the electron transfer, which would be helpful for improving the overall catalytic activity. The long-term stability through continuous CV sweep and chronopotentiometric measurements were carried out in 1.0 M KOH. Impressively, there was only an increase of 8 mV in the overpotential at 50 mA cm$^{-2}$ after 47000 cycles (Fig. 6c), with its morphology retained (Supplementary Fig. 14), indicating excellent long-term durability. In a control experiment, the pristine ICLDH/NF exhibited a significant decrease (ca. 39%) in catalytic current density at 1.6 V vs. RHE after only 3000 cycles. The Faradaic efficiency of e-ICLDH@GDY/NF for OER is close to 100% (97.98 ± 0.81%, Supplementary Fig. 15). Thus, the presence of GDY played an important role in improving the OER activity and stability.

We examined the relationship between temperature relation and electrocatalytic activity to determine the kinetic barriers for the OER (Fig. 6d)[48]. At elevated temperatures, the OER proceeded more rapidly. The overpotentials required to deliver a current density of 10 mA cm$^{-2}$ decreased from 216 mV (at 25 °C) to 191 mV (at 50 °C), 175 mV (at 60 °C), and 158 mV (at 70 °C). As displayed in Fig. 6e, the Tafel slopes decreased from 43.6 mV dec$^{-1}$ (at 25 °C) to 40.7 mV dec$^{-1}$ (at 50 °C), 37.6 mV dec$^{-1}$ (at 60 °C) and 27.5 mV dec$^{-1}$ (at 70 °C). The estimated electrochemical activation energies ($E_a$) of the OER can be extracted from the Arrhenius plots (Fig. 6f), according to the Eq. (1):

$$E_a = -2.303R \left[ \frac{\partial \log(j)}{\partial \left( \frac{1}{T} \right)} \right] \quad (1)$$

where $R$ is the universal gas constant and $T$ is absolute temperature. We constructed Arrhenius plots based on the current densities recorded at different potentials. As the potentials increased from 1.43 to 1.44 to 1.45 V (v. RHE), the activation energies decreased from 100.9 to 91.52 to 82.33 kJ mol$^{-1}$, respectively, indicating the potential-determining step of our electrocatalyst.

**Electrocatalytic HER catalysis**. We investigated the HER performance of our structures ($2H_2O + 2e^- \rightarrow H_2 + 2OH^-$) in 1.0 M KOH at 1 mV s$^{-1}$. Figure 6g and Supplementary Fig. 16 provide the polarization curves after $iR$-compensation of the as-synthesized samples. e-ICLDH@GDY/NF exhibited the lowest overpotentials of 43, 174, 215, and 256 mV at 10, 50, 100, and 1000 mA cm$^{-2}$, respectively. These values are better than those of the pristine ICLDH/NF (202 mV at 10 mA cm$^{-2}$), monometallic LDHs (Fig. 6g), and most previously reported nonprecious electrocatalysts in alkaline medium (Supplementary Table 2). Similar to that observed in OER (Supplementary Fig. 10), the HER results (Supplementary Fig. 16) also showed that both the GDY introducing and Fe incorporation into CLDH contributed to the improvement of the HER activity, and Fe incorporation

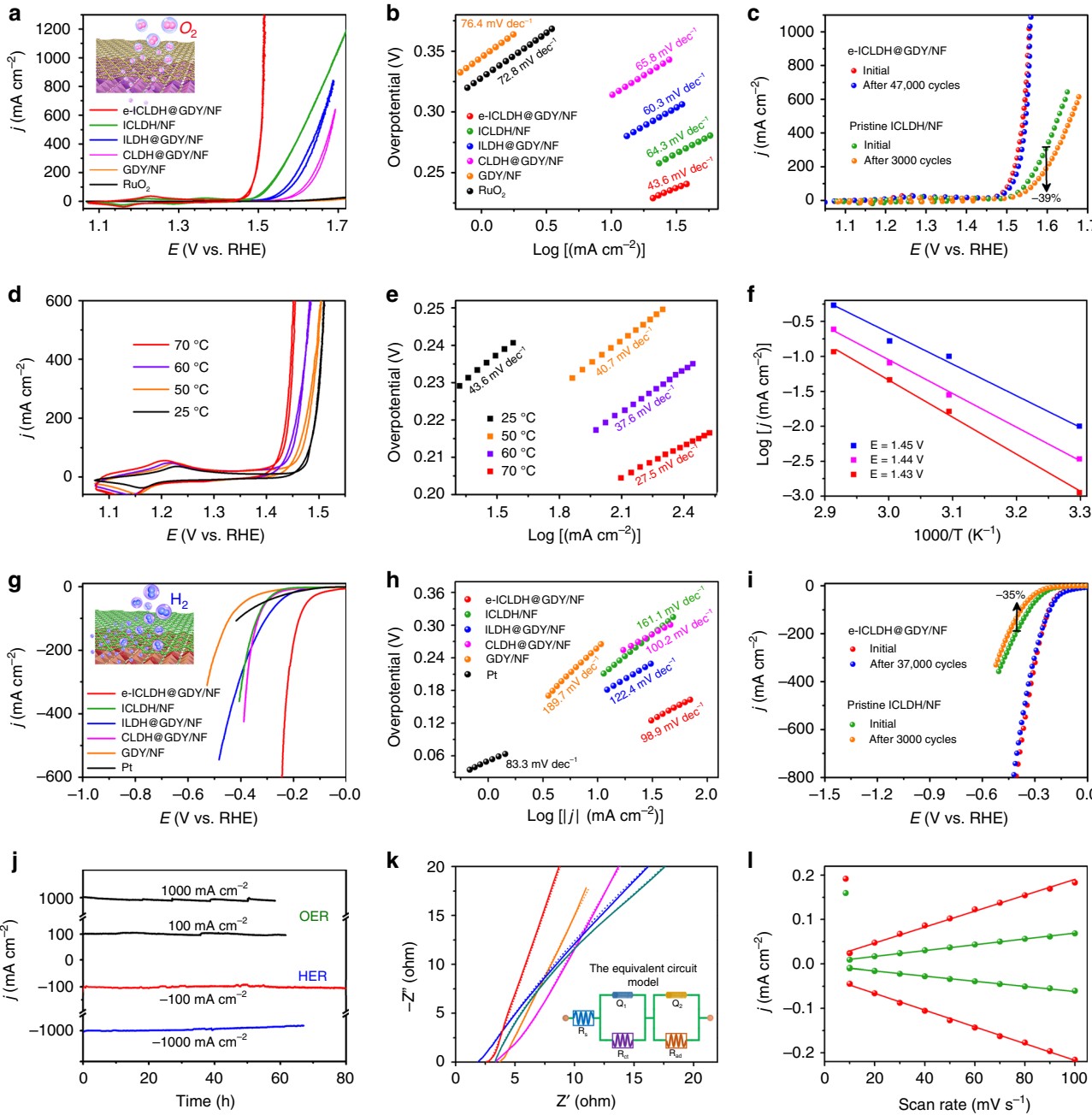

**Fig. 6** Electrocatalytic performance of e-ICLDH@GDY/NF. **a** OER CV curves and **b** corresponding Tafel plots of as-synthesized samples. **c** Polarization curves of e-ICLDH@GDY/NF and pristine ICLDH/NF, recorded before and after performing 47,000 and 3000 cycles, respectively, of the OER. **d** OER CV curves and **e** corresponding Tafel plots for e-ICLDH@GDY/NF, recorded at various temperatures. **f** Arrhenius plots for the OER performed over e-ICLDH@GDY/NF at various potentials. **g** HER polarization curves and **h** corresponding Tafel plots for the as-synthesized samples. **i** HER polarization curves of e-ICLDH@GDY/NF and pristine ICLDH/NF, recorded before and after 37,000 and 3000 cycles, respectively, of the HER. **j** Chronopotentiometric curves of e-ICLDH@GDY/NF recorded at 100 and 1000 mA cm⁻² for both the OER and HER. **k** Experimental (solid line) and fitted (dashed line) Nyquist plots for e-ICLDH@GDY/NF (red), ICLDH/NF (green), ILDH/NF (blue), CLDH/NF (magenta), and GDY/NF (orange), respectively; inset: equivalent circuit model. **l** Capacitive current densities at 0.818 V, plotted with respect to the scan rate, for e-ICLDH@GDY/NF (red) and ICLDH/NF (green)

contributes more. The small Tafel slope value (98.9 mV dec⁻¹), very close to that of Pt (83.3 mV dec⁻¹), confirmed the excellent HER activity of e-ICLDH@GDY/NF (Fig. 6h, Supplementary Fig. 17). We performed continuous cycling tests at 100 mV s⁻¹ to evaluate the durability of e-ICLDH@GDY/NF during the HER process. Only a negligible change in current density occurred after 37,000 continuous CV cycles (Fig. 6i). In contrast, the pure ICLDH/NF exhibited a great decrease in current density, from 186.5 to 120.6 mA cm⁻², at –0.4 V (vs. RHE) after only

3000 cycles. The elemental composition and morphology of e-ICLDH@GDY/NF were both preserved well after the long-term stability tests (Supplementary Fig. 18). We also examined the long-term electrocatalytic stability of the e-ICLDH@GDY/NF structure at current densities of 100 and 1000 mA cm⁻² for both the OER and HER. During the continuous OER test, the current densities of e-ICLDH@GDY/NF remained nearly constant for 60 and 58 h at the larger current densities of 100 and 1000 mA cm⁻², respectively (Fig. 6j). The excellent stability of e-ICLDH@GDY/

NF for the HER was further evidenced through constant-current electrolysis tests performed at current densities of 100 mA cm$^{-2}$ for 80 h (no change in current density) and 1000 mA cm$^{-2}$ for 67 h (Fig. 6j). The Faradaic efficiency of e-ICLDH@GDY/NF for HER was determined to be 98.96 ± 0.86% (Supplementary Fig. 19). Taken together, these findings confirmed the outstanding stability and superiority of the e-ICLDH@GDY/NF electrode.

To determine the origin of the excellent activity of e-ICLDH@GDY/NF, we recorded electrochemical impedance spectroscopy (EIS) Nyquist plots (Fig. 6k, Supplementary Fig. 20) to study its charge transfer behavior[39]. We employed an equivalent circuit model (inset to Fig. 6k) featuring solution resistance ($R_s$), charge transfer resistance ($R_{ct}$), and adsorption resistance ($R_{ad}$) to fit the Nyquist plots. e-ICLDH@GDY/NF possessed the lowest values of $R_s$ and $R_{ct}$ (2.46 and 1.42 Ω, respectively) among all the electrocatalysts (Supplementary Table 3); in particular, these values were much lower than those of the pristine ICLDH/NF ($R_s$ = 3.45 Ω; $R_{ct}$ = 21.55 Ω) and GDY/NF ($R_s$ = 3.78 Ω; $R_{ct}$ = 23.63 Ω), indicating that e-ICLDH@GDY/NF had the most favorable charge transfer kinetics. In addition, the electrochemical measurements revealed that e-ICLDH@GDY/NF had a capacitance of 2.0 mF cm$^{-2}$, approximately 3.3 times that of the pristine ICLDH/NF (0.6 mF cm$^{-2}$, Fig. 6l,

Supplementary Fig. 21), suggesting that the former had relatively larger electrochemical surface area (ECSA). Turnover frequencies (TOFs) have also been used to assess the information about the intrinsic activity (see Supplementary Methods for details). The e-ICLDH@GDY/NF shows TOF values 8.44 s$^{-1}$ at 200 mV for HER and 2.34 s$^{-1}$ at 250 mV for OER, respectively, which are larger than that of pure ICLDH and most of recently reported catalysts (Supplementary Table 4). The improvement in the catalytic activity of e-ICLDH@GDY/NF was presumably due to the incorporation of ICLDH with GDY, which significantly facilitated electron transport and increased the number of electrocatalytic active sites.

**Overall water splitting**. Considering the outstanding HER and OER performance and stability of the bifunctional e-ICLDH@GDY/NF electrocatalyst, we assembled an alkaline electrolyzer, using e-ICLDH@GDY/NF as both the anode and cathode, in 1.0 M KOH (Fig. 7a). As shown in Fig. 7b, c, the e-ICLDH@GDY/NF electrode exhibited the best OWS activity among all samples. For example, the CV curves in Fig. 7c (with iR-correction) show that the e-ICLDH@GDY/NF electrode exhibited OWS activity—with small cell voltages of 1.43, 1.46, and

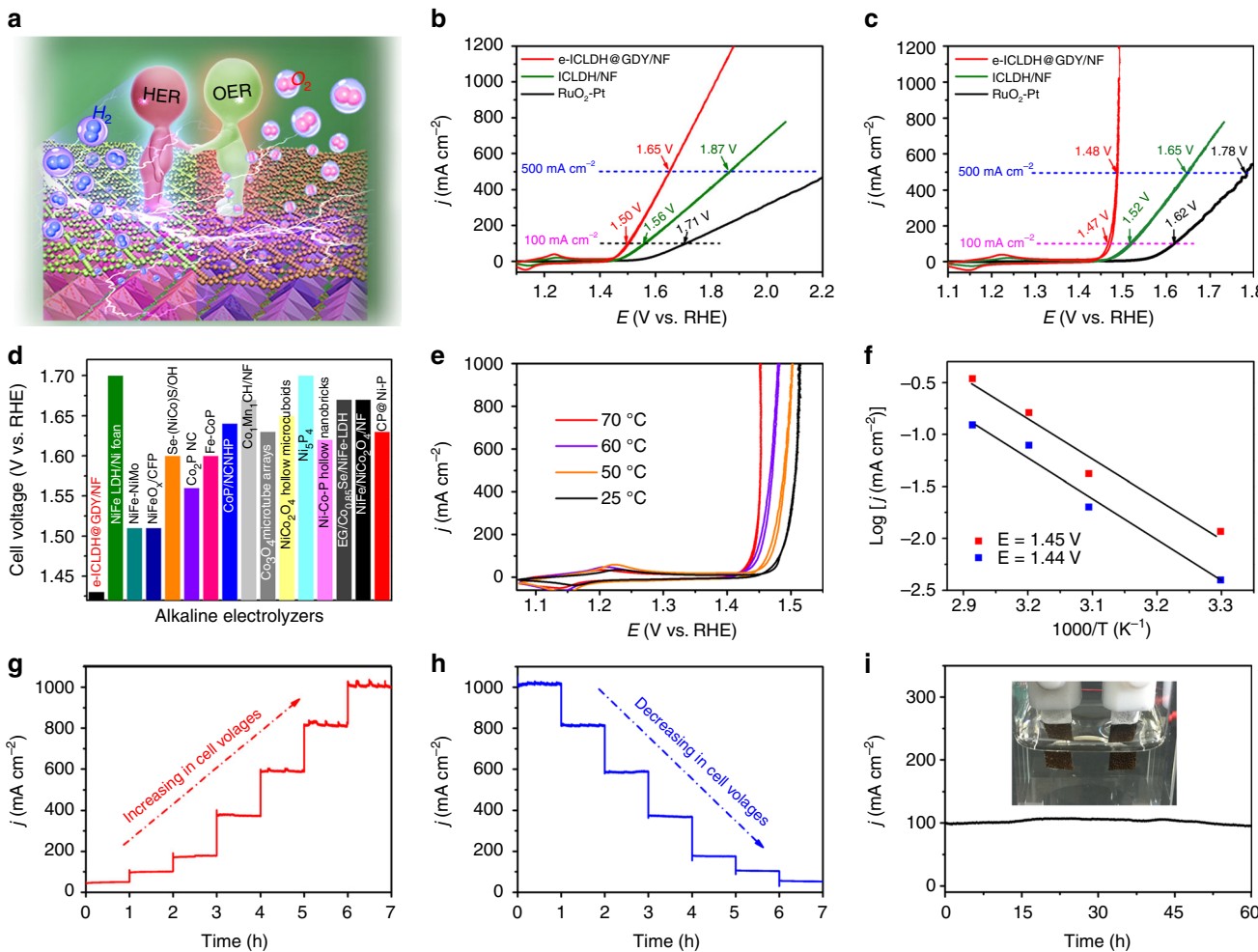

**Fig. 7** OWS performance of the electrocatalyst. **a** Schematic representation of the OWS process. **b, c** OWS activities of e-ICLDH@GDY, a precious RuO$_2$-Pt couple, and the pristine ICLDH/NF **b** without and **c** with iR-correction. **d** OWS activities of e-ICLDH@GDY and previously reported electrocatalysts. **e** CV curves for OWS processes performed at various temperatures. **f** Arrhenius plots for OWS processes performed on e-ICLDH@GDY/NF at various potentials. **g, h** Current density–time curves measured with **g** increasing (red line) and **h** decreasing (blue line) cell voltages. **i** Current density of the e-ICLDH@GDY/NF electrode in an alkaline electrolyzer under a constant cell voltage of 1.56 V, measured over 60 h (inset: the photograph for the two-electrode system)

1.49 V to deliver 10, 100, and 1000 mA cm$^{-2}$, respectively—that was much better than those of ICLDH/NF (1.47 and 1.52 V at 10 and 100 mA cm$^{-2}$, respectively) and a RuO$_2$–Pt couple (1.53 and 1.62 V at 10 and 100 mA cm$^{-2}$, respectively), as well as those of other reported state-of-the-art electrodes, including Co$_1$Mn$_1$CH (1.68 V at 10 mA cm$^{-2}$)[10], NiFeLDH/NF (1.70 V at 10 mA cm$^{-2}$)[49], and P-NiFe (1.51 V at 10 mA cm$^{-2}$)[50] (Fig. 7d, Supplementary Table 5). The OWS reaction can be driven by a single-cell AA battery with a nominal voltage of 1.5 V (Supplementary Fig. 22, Supplementary Movie 1, 2).

Next, we examined the relationship between the temperature and the OWS activity of e-ICLDH@GDY/NF. As displayed in Fig. 7e, the cell voltage required to deliver a current density of 10 mA cm$^{-2}$ decreased significantly from 1.43 to 1.41 V as the temperature increased from 25 to 70 °C. We extracted the kinetic barriers involved in the OWS process from the Arrhenius plots (Fig. 7f). The values of $E_a$ for the OWS processes performed at cell voltages of 1.44 and 1.45 V (vs. RHE) were 77.22 and 74.34 kJ mol$^{-1}$, respectively, revealing that the activation energy for the OWS process decreased upon increasing the cell voltage.

The water splitting capability of our electrolyzer was then examined at various cell voltages (Fig. 7g, h). Upon increasing the applied voltages, the current densities increased from approximately 50 mA cm$^{-2}$ to approximately 1000 mA cm$^{-2}$ and stabilized rapidly. The current densities decreased back from 1000 to 50 mA cm$^{-2}$ upon decreasing the applied voltage, revealing the system's excellent stability. We attribute the small current fluctuations on the curves at large current densities to the rapid or vigorous evolution of gas bubbles from the electrode. We examined the long-term stability of the e-ICLDH@GDY/NF–based electrolyzer under a constant cell voltage of 1.56 V. Figure 7i reveals that the current density of 100 mA cm$^{-2}$ remained constant for more than 60 h. The Faradaic efficiency of O$_2$ evolution on e-ICLDH@GDY/NF was 97.40 ± 1.30% (Supplementary Fig. 23). We observed vigorous evolution of gases from the electrode (inset in Fig. 7i), but no obvious fluctuations in the current density–time curve. Most impressively, e-LDH@GDY/NF required only 1.48 and 1.49 V to achieve stable current densities of 500 and 1000 mA cm$^{-2}$, respectively, indicating that it is a very favorable electrocatalyst for rapid water splitting with possible practical applications.

We have prepared an active and stable bifunctional HER/OER electrocatalyst, featuring GDY sandwiched between exfoliated ICLDH nanosheet arrays, through a GDY-induced intercalation-exfoliation method. This bifunctional electrocatalyst provided low OER overpotentials of 216, 249, and 278 mV to reach current densities of 10, 100, and 1000 mA cm$^{-2}$, respectively, and HER overpotentials of 43, 215, and 256 mV to achieve current densities of 10, 100, and 1000 mA cm$^{-2}$, respectively, in 1.0 M KOH. It also displayed outstanding long-term stability, even when operated at a large current density of 1000 mA cm$^{-2}$ for both the OER (>58 h) and HER (>67 h). Impressively, when assembled into an alkaline electrolyzer, the electrocatalyst provided low voltages of 1.43, 1.46, and 1.49 V at current densities of 10, 100, and 1000 mA cm$^{-2}$, respectively, for the OWS. Both experimental and theoretical data revealed that the incorporation of ICLDH with GDY improved the charge transport kinetics of the catalyst. Experimental results also confirmed that GDY played critical roles in enhancing the electrocatalytic activity and long-term stability of the electrocatalyst. Our synthetic strategy appears to be a simple and efficient method for in situ preparation of exfoliated LDH nanosheet arrays, without the need for additional binders.

## Methods

The calculation details are given in the Supplementary Methods.

**Synthesis of e-ICLDH@GDY/NF.** ICLDH/NF was initially prepared via a hydrothermal method. Briefly, cobalt nitrate (1 mmol), urea (5 mmol), ferrous chloride (0.25 mmol) and ammonium fluoride (2 mmol) were added to a mixed solution of H$_2$O (28 mL) and ethanol (5 mL) under stirring for 30 min. The mixture was then shifted to a 50 mL Teflon lined autoclave containing a piece of Ni foam (2 cm × 2 cm) at 120 °C for 8 h. After 8 h, the ICLDH/NF samples were taken out and washed with deionized water thoroughly and dried at 40 °C in a vacuum oven. The obtained ICLDH/NF together with several pieces of Cu foils (2.0 × 2.0 cm) were added to a three-necked bottle containing acetone/pyridine/tetra-methylethylenediamine (100:5:1). HEB (0.07 mmol) dissolved in pyridine (50 mL) was added very slowly into the above mixture. After a reaction at 50 °C for 25 h under Ar atmosphere, the e-ICLDH@GDY/NF was obtained. All products was washed with 2 M HCl, 2 M NaOH, acetone, DMF and deionized water subsequently before use.

**Characterization.** The X-ray photoelectron spectroscopy (XPS, Thermo Scientific ESCALab 250Xi), FTIR spectroscopy (Thermo Fisher Nicolet6700), and Raman spectroscopy (Renishaw-2000) were used to get the structural information of samples. Field-emission SEM (Hitachi S-4800) and TEM, high-resolution TEM (JEM-2100F) were used to characterize the morphologies of the samples.

**Electrochemical measurements.** Electrochemical measurements were performed on a typical three-electrode system (CHI 760E, CH Instruments, Inc. Shanghai) using a saturated calomel electrode (SCE) as reference electrode, graphite rod as counter electrode, and the as-synthesized electrocatalyst as the working electrode. OER CV curves and HER polarization curves were recorded at 1 mV s$^{-1}$ in O$_2$-saturated and H$_2$-saturated 1.0 M KOH, respectively. Continuous cycling tests were carried out at 100 mV s$^{-1}$. EIS data were collected at open-circuit potentials (frequency range: 100 KHz to 0.1 Hz) and fitted by an equivalent circuit [R(QR)(QR)]. All potentials reported have been converted to the reversible hydrogen electrode (RHE, $E_{RHE} = E_{SCE} + E^0_{(SCE)} + 0.059$ pH).

## Data availability

The data that support the plots within this paper and other finding of this study are available from the corresponding author upon reasonable request.

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

## Acknowledgements

This work was supported by the National Nature Science Foundation of China (21790050 and 21790051), the National Key Research and Development Project of China (2016YFA0200104), and the Key Program of the Chinese Academy of Sciences (QYZDY-SSW-SLH015).

## Author contributions

Yuliang Li conceived and designed the research and analysis, reviewed and edited this manuscript. L.H. carried out the catalyst synthesis, electrochemical experiments and wrote corresponding discussions. Y.X. helped to collect, analyze and organize the data, and write the draft. B.H. gave important help in theoretical calculations. H.Y., C.Z., D.Z., D.J., Y.Z., Yongjun Li and H.L. gave helpful advice.

## Additional information

**Competing interests:** The authors declare no competing interests.

