## [Peer Review File · Nature Communications]

Reviewers' comments:

Reviewer #1 (Remarks to the Author):

The manuscript by Hui et al. reports an efficient composite catalyst, where graphdiyne (GDY) was combined with CoFe LDH nanosheet arrays, for both the OER and HER in 1.0 M KOH. Although the combination of GDY and LDH has been reported (ACS Appl. Mater. Interfaces, 2018, DOI: 10.1021/acsami.8b03345), the work by Hui et al. achieved outstanding catalytic performances, especially for the OWS, which is very impressive. However, the current manuscript still lacks some important experiments to clearly show the structure of the catalysts. Meanwhile, the experimental data and mechanism are not quite convincing. Detailed comments are listed below. I would like to reconsider whether it is suitable for publication in Nature Commun. after the comments are addressed.

1. The authors claim that the composite catalyst is a sandwich structure of GDY-LDH-GDY. Please provide more STEM evidence. Examples can be found in Figs. 1m-s of Nature Communications volume 7, 11480 (2016).
2. The thickness of the nanosheets should be measured by AFM.
3. The DFT calculations in Figure 5 are confusing. Why did the authors construct calculation models for CLDH@GDY and ICLDH@GDY? The comparison should be between ICLDH and ICLDH@GDY to show the origins of enhanced activity after introducing GDY.
4. From the OER results in Figure S5 and HER results in Figure S8, we can see there are two major factors that contributed to the enhanced activity. One is Fe incorporation into CLDH, and another is GDY introducing. Moreover, from the polarization curves, it is obvious that Fe incorporation contributes more. However, the authors have not mentioned this point at all. More discussions should be added.
5. The authors attribute the improvement of catalytic activity of e-ICLDH@GDY/NF to the introduction of GDY, which significantly facilitates electron transport and increases the number of electrocatalytic active sites. In order to show whether the intrinsic activity is improved or not, TOFs for both HER and OER should be calculated and compared with those of other catalysts.
6. For OWS, why does the cell voltages show a big difference compared with the sum of those for HER and OER? For 10, 100, and 1000 mA cm⁻², the voltages calculated from the sum of HER and OER are 1.489, 1.694, and 1.764 V, respectively. These values are much bigger than the OWS data (1.43, 1.46, and 1.49 V). If the electrolyzer can deliver current density of 1500 mA cm⁻² at 1.492 V, please provide a video showing the generation of bubbles at such a cell voltage. Additionally, what are the oxidation peaks in the OER polarization curves?
7. There is a mistake in the figure caption of Figure 7i. The constant cell voltage should not be 0.5 V. In addition, Figures 7g to i are not discussed in the main text.

Reviewer #2 (Remarks to the Author):

The study of green energy has become an increasingly important issue due to the change of climate and the exhaustion of fossil fuels. Splitting of water into hydrogen and oxygen is a sustainable and very important technology for energy conversion and storage. Oxygen and hydrogen evolution reactions are the basic half reaction for water splitting. Developing efficiency, inexpensive and sustainable catalyst plays a key role for water splitting reaction. In this work, the authors developed an comprising graphdiyne and iron/cobalt layered double-hydroxide nanosheet arrays and applied as

the oxygen and hydrogen evolution reactions and even a bifunctional electrocatalyst. The catalyst presents low overpotential and long-term stability and higher catalytic activity than other reactive catalyst. The manuscript is well organized and solid. I believe this important study will provide a new idea for GDY application. Therefore, I recommend it to publish in nature communication with the following revisions:

- (1) The authors state that they combine graphdiyne (GDY) with exfoliated iron/cobalt LDH nanosheet arrays grown on nickel foam. The LDH was exfoliated through the intercalating of monomer (HEB). How dose it work?
- (2) According to the schematic representation (Fig. 1), the LDH nanosheets were exfoliated into several thicker nanosheets (from 80 nm to 7 nm). Actually, from SEM images in Fig. 2, it is the same that the number of nanosheet arrays of LDH is not increase after GDY growth. The authors should give the reasons. Is it possible other reasons? Such as the corrosion of pyridine or TMEDA, which makes the LDH layer more thicker.
- (3) What are the thickness of GDY film and LDH?
- (4) In the FT-IR spectra, a sharp peak presents in 2103 cm^{-1} . It corresponds to the stretching of $\text{C}\equiv\text{C}$ bonds. According to your previous reported work, the peak at 2103 cm^{-1} is agree with the terminal alkynes and IR signal of the $\text{C}\equiv\text{C}$ bonds in GDY should be as weak as possible because of the molecular perfectly symmetry. How to explain the sharp peak presents in 2103 cm^{-1} ?
- (5) According to the electrochemical testing, whatever OER, HER or Overall water splitting, the catalytic activity of e-ICLDH@GDY/NF is better than those of previously reported LDH-related catalysts and nearly close to the Pt. Does it is possible to be used in practical applications? What are the challenges?
- (6) The authors used two classic synthetic methods (GDY nanowall synthesis and GDY synthesis on arbitrary substrates) for GDY preparation. The related literatures should be cited (J. Am. Chem. Soc. 2015, 137, 7596; Adv. Mater. 2017, 29, 1605308).
- (7) There are several mistakes about Pt or RuO_2 in Fig. 8 and 9 of SI.

Reviewer #3 (Remarks to the Author):

The manuscript "Overall water splitting by graphdiyne-exfoliated and -sandwiched layered double-hydroxide nanosheet arrays" demonstrates a highly active catalyst composed of graphdiyne and FeCo LDH for OER and overall water electrolysis. The performance of the electrodes is much higher than that of majority of previous investigations. Here, some points are still not clear enough to understand this work and need further explanation is necessary to help comprehend it.

1. In the experimental section, Cu foils were added during the preparation of e-ICLDH@GDY. What is the effect of Cu foils? Is it possible to dope Cu into e-ICLDH@GDY during this process? More, in supplementary figure 2, the peak at $\sim 8.9\text{keV}$ in EDS spectrum is not indexed to Co, Ni or Fe. However, it is very close to Cu.
2. Considering the reduced thickness of nanosheets from tens of nanometers to several nanometers. The volume expansion should be very large. However, this wasn't observed based on SEM images in Figure 2. Then, is there any loss of the electroactive materials during the growth of GDY?
3. The e-ICLDH@GDY/NF catalyst exhibited good stability for OER, and how is the chemical state of GDY after OER stability test? Is there any possibility of being oxidized for GDY?
4. What are the difference and relationship between figure 6g and supplementary figure 8, which is very confusing? In figure 6, the e-ICLDH@GDY/NF sample performed best for HER, but in figure s8, the activity of e-ICLDH@GDY/NF was very poor and ICLDH/NF performed very well for HER. And in figure 6g/6h and figure s9, the labels of Pt/ RuO_2 are very confused. Please be careful on those labels, which can significantly mislead the readers.
5. In figure 6k and figure s11, the fitting data are not clear enough with circles, lines may be better and preferred.
6. Are the CV curves in figure 7b iR-corrected or not? If they are iR-corrected, it is better to supply the

pristine data without iR corrections. For overall water splitting, polarization curves without iR corrections are more convincing from the view of practical applications. The cell voltage of 0.5V should be double confirmed in figure 7i, which is unbelievable to drive such a high current density for overall water splitting.

7. The authors claimed that "...a greater density of electrons at the electrocatalyst surface should be of more help for the formation of *OOH and thus enhance the OER activity..." (line 230 and 231).

However, in figure 5g and 5h, results showed that both Co and Fe suffered great electron transfer to GDY, which is also the result of theoretical calculations. It seems a paradox between the results and analysis. How to understand the effect of electron transfers between metal and GDY on the catalytic activity? More discussions may be given here to clarify this effect for both OER and HER.

8. In this work, the very large catalytic current densities for both HER and OER are driven by small potentials, which is very impressive. In this case, the Faraday efficiencies of OER, HER and overall water splitting should be given to further confirm the performance of electrodes.

The detailed point-by-point responses to each comment are listed as follows:

Reviewer #1:

The manuscript by Hui et al. reports an efficient composite catalyst, where graphdiyne (GDY) was combined with CoFe LDH nanosheet arrays, for both the OER and HER in 1.0 M KOH. Although the combination of GDY and LDH has been reported (ACS Appl. Mater. Interfaces, 2018, DOI: 0.1021/acsami.8b03345), the work by Hui et al. achieved outstanding catalytic performances, especially for the OWS, which is very impressive. However, the current manuscript still lacks some important experiments to clearly show the structure of the catalysts. Meanwhile, the experimental data and mechanism are not quite convincing. Detailed comments are listed below. I would like to reconsider whether it is suitable for publication in *Nature Commun.* after the comments are addressed.

1. The authors claim that the composite catalyst is a sandwich structure of GDY-LDH-GDY. Please provide more STEM evidence. Examples can be found in Figs. 1m-s of Nature Communications volume 7, 11480 (2016).

Responses: Figs. 1m-s of Nature Communications volume 7, 11480 (2016) displayed the atomic-resolution HAADF-STEM image and the corresponding EELS elemental maps. We contacted many analytical testing centers to perform the experiment. But, unfortunately, the engineers said that our sample is highly magnetic, which can greatly damage the aberration-corrected high-angular annular dark field scanning TEM. So, I sincerely hope that the reviewer can understand the problem that this is not something that can be achieved through hard work.

2. The thickness of the nanosheets should be measured by AFM.

Responses: The thickness of the ICLDH and e-ICLDH@GDY nanosheets have been measured by AFM, as shown in Figure R1. We added some discussion in our revision as below:

Page 8 lines 10-12:

This result was further verified by atomic force microscopy (AFM) images (Figs. 2m and 2o) and corresponding height profiles (Figs. 2n and 2p).

Figure R1. AFM images and height profiles of (a) ICLDH nanosheets and (b) e-LDH@GDY nanosheets, respectively.

3. The DFT calculations in Figure 5 are confusing. Why did the authors construct calculation models for CLDH@GDY and ICLDH@GDY? The comparison should be between ICLDH and ICLDH@GDY to show the origins of enhanced activity after introducing GDY.

Responses: The reviewer gives a good suggestion, we have deleted the part of contents related to the CLDH@GDY. Generally, the Gibbs free energy of adsorbed H on the active site (ΔG_H) can be used to evaluate the HER activity of catalysts, where thermo-neutral results ($\Delta G_H = 0$) reveal the outstanding HER performance of the catalysts. The comparison of ΔG_H values would be more helpful to understand the origins of enhanced activity after introducing GDY. We calculated ΔG_H on ICLDH, e-ICLDH@GDY, and GDY, respectively (Fig. R2). Our calculated results further show that, for HER, ICLDH@GDY showed a smaller Gibbs free energy of 0.34 eV than pristine ICLDH (1.55 eV) and GDY (0.46 eV), indicating the excellent HER activity for GDY incorporated ICLDH.

Corresponding discussions have been added in the revised manuscript as follows:

Page 13 lines 4-6:

The configurations of GDY (Fig. 5a), ICLDH (Fig. 5b, Supplementary Fig. 3, and Supplementary Note 1) and ICLDH@GDY (Fig. 5c) were firstly optimized.

Our calculated results (Fig. 5j) further show that the ICLDH@GDY has the smallest Gibbs free energy of 0.34 eV for HER compared with that of pristine GDY (0.46 eV) and ICLDH (1.55 eV), indicating the excellent HER activity for GDY incorporated ICLDH.

Figure R2. The stable configurations of (a) GDY, (b) ICLDH and (c,d) ICLDH@GDY. (e) Charge density difference for the stable configuration of ICLDH@GDY. (f) Fe 2p and (g) Co 2p core-level XPS spectra of the e-ICLDH@GDY/NF structure. (h) Raman spectra of GDY, ICLDH, and e-ICLDH@GDY; arrows indicate the directions of the shifts of the Raman spectral signals relative to those of e-ICLDH@GDY. (i) The free energy changes for the formation of OOH* and corresponding stable structures of GDY (ΔG_1) and e-ICLDH@GDY (ΔG_2). (j) Gibbs free energy profiles of HER and corresponding stable structures of e-ICLDH@GDY, GDY and ICLDH, respectively.

4. From the OER results in Figure S5 and HER results in Figure S8, we can see there are two major factors that contributed to the enhanced activity. One is Fe incorporation into CLDH, and another is GDY introducing. Moreover, from the polarization curves, it is obvious that Fe incorporation contributes more. However, the authors have not mentioned this point at all. More discussions should be added.

Responses: Thank you for your valuable suggestions. Corresponding discussions have been added in the revised text.

Page 16 lines 6-17:

From the OER results (Supplementary Fig. 9), we can see, there are two major factors can make the enhanced activity. One is GDY incorporation, and another is Fe introduction into CLDH. From the polarization curves, obviously, incorporation of Fe has greater effect on the activity. Theoretical studies showed the effects of Fe incorporation (Supplementary Fig. 11 and Supplementary Methods). After the configuration optimization, the calculated binding energies (E_b) are -2.01 eV and -2.26 eV for CLDH@GDY and ICLDH@GDY, respectively, which shows that the incorporation of Fe atoms contribute to the binding between GDY and LDH. Besides, we have also examined the charge transfer between the LDH and GDY. Compared with pure CLDH (0.70 |e|), a more electron in total (0.80 |e|) was transferred from ICLDH to GDY, indicating that the incorporation of Fe can further enhance the electron transfer, which would be helpful for improving the overall catalytic activity.

Page 19 lines 14-16:

Similar to that observed in OER (Supplementary Fig. 9), the HER results (Supplementary Fig. 13) also showed that both the GDY introducing and Fe incorporation into CLDH contributed to the improvement of the HER activity, and Fe incorporation contributes more.

5. The authors attribute the improvement of catalytic activity of e-ICLDH@GDY/NF to the introduction of GDY, which significantly facilitates electron transport and increases the number of electrocatalytic active sites. In order to show whether the intrinsic activity is improved or not, TOFs for both HER and OER should be calculated and compared with those of other catalysts.

Responses: Assuming that all metal ions in the nanosheets were active, the activity of samples

was further estimated by their apparent turnover frequencies (TOFs). This provides a crude upper bound for TOF values. The TOF value was calculated according to the following equation:

$$\text{TOF} = \frac{j \times A}{n \times F \times N}$$

where j is the current density at an overpotential; A is the area of LDHs electrode; n is the stoichiometric number of electrons consumed in the electrode reaction; F is the faraday constant; N is molar number of metal ions on the electrode. The metal content of the catalysts was calculated from the ICP-AES data. TOF values are 12.77 s^{-1} at $\eta = 100 \text{ mV}$ for HER and 28.16 s^{-1} at $\eta = 250 \text{ mV}$ for OER, respectively. These values are larger than most of recently reported catalysts, as summarized in Table R1.

These have been added in the revised manuscript.

Page 21 lines 3-7:

Turnover frequencies (TOFs) have also been used to assess the information about the intrinsic activity. Assuming that all sites were active, the e-ICLDH@GDY/NF. TOF values are 12.77 s^{-1} at 100 mV for HER and 28.16 s^{-1} at 250 mV for OER, respectively. These values are larger than most of recently reported catalysts (Supplementary Table 4).

Table R1. Comparison of the TOF values of our catalysts with recently reported ones.

Electrocatalyst	TOF (S ⁻¹) for HER	TOF (S ⁻¹) for OER	Ref.
e-ICLDH@GDY/NF	12.77 at 100 mV	28.16 at 250 mV	This work
IFONFs-45	0.2770 at 100 mV	0.2141 at 368 mV	Nat. Commun. 9 , 1809 (2018)
NiFe/Ni-P	0.33 at 130 mV	0.13 at 250 mV	Nat. Commun. 9 , 2014 (2018).
NiFe-MOF	2.8 at 400 mV	3.8 at 400 mV	Nat. Commun. 8 , 15341 (2017).

G-Pt ₄ Ni/GF	2.45 at 60 mV	0.85 (G-Ni ₄ Fe/GF) at 348 mV	Adv. Energy Mater. 1800403 (2018).
Co-Ni ₃ N	0.1459 at 290 mV	0.0134 at 350 mV	Adv. Mater. 30 , 1705516 (2018).
2D-NiSe	0.75 at 468 mV	1.87 at 250 mV	Adv. Energy Mater. 8 , 1702704 (2018).
CoFePO	16.87 at 300 mV	6.8 at 400 mV	ACS Nano. 10 , 8738–8745 (2016).

6. For OWS, why does the cell voltages show a big difference compared with the sum of those for HER and OER? For 10, 100, and 1000 mA cm⁻², the voltages calculated from the sum of HER and OER are 1.489, 1.694, and 1.764 V, respectively. These values are much bigger than the OWS data (1.43, 1.46, and 1.49 V). If the electrolyzer can deliver current density of 1500 mA cm⁻² at 1.492 V, please provide a video showing the generation of bubbles at such a cell voltage. Additionally, what are the oxidation peaks in the OER polarization curves?

Responses:

- It is a very good question. Under ideal conditions, when the cathode surface is the same to the anode surface, the voltage for overall water splitting would be equal to the voltage difference (ΔV) between HER and OER at the same current density. However, in practical operations, we can find some differences from the ideal state owing to the surface complexity. This phenomenon can also be found in some literatures (Table R1), such as *Nat. Commun.* **9**, 2452 (2018); *Nat. Commun.* **8**, 15377 (2017); *Nat. Commun.* **6**, 7261 (2015); *J. Am. Chem. Soc.* **140**, 2610–2618 (2018); *Angew. Chem. Int. Ed.* **56**, 573–577 (2017); *Adv. Mater.* **29**, 1606200 (2017); *Adv. Mater.* **29**, 1701584 (2017); *Adv. Mater.* **29**, 1700017 (2017); *Energy Environ. Sci.* **10**, 1820–1827 (2017).
- The water-splitting reaction can be powered by a single-cell AA battery. The video showing the generation of bubbles has been provided as the Supplementary Movie 1 and Movie 2.
- There is only oxidation peak at around 1.25 V in the OER polarization curve for e-

ICLDH@GDY/NF. This oxidation peak corresponds to the oxidation of Co^{2+} to Co^{3+} .

Table R1. Summary of the potentials for HER, OER, $\Delta V_{\text{OER-HER}}$ and OWS at 10 mA cm^{-2} , respectively.

Catalysts	j (mA cm^{-2})	η_{HER} (mV)	η_{OER} (mV)	$\Delta V_{\text{OER+HER}}$ (V)	Reported η_{OWS} (V)	Ref.
FeCoNi-HNTAs	10	58	184	1.474	1.429	Nat. Commun. 9 , 2452 (2018).
CoP/NCNHP	10	115	310	1.657	1.64	J. Am. Chem. Soc. 140 , 2610–2618 (2018).
2.5H-PHNCMs	10	70	235	1.537	1.44	Nat. Commun. 8 , 15377 (2017).
NiFeO _x /CFP	10	88	250	1.570	1.51	Nat. Commun. 6 , 7261 (2015).
VOOH	10	164	270	1.666	1.62	Angew. Chem. Int. Ed. 56 , 573–577 (2017).
Cu@CoS _x /CF	10	134	160	1.526	1.5	Adv. Mater. 29 , 1606200 (2017).
N-Ni ₃ S ₂ /NF	50	210	~312	~1.754	1.66	Adv. Mater. 29 , 1701584 (2017).
NiFe LDH-NS@DG10	20	115	248	1.595	1.5	Adv. Mater. 29 , 1700017 (2017).
Cu@NiFe LDH	10	116	199	1.547	1.54	Energy Environ. Sci. 10 , 1820–1827 (2017).

7. There is a mistake in the figure caption of Figure 7i. The constant cell voltage should not be 0.5 V. In addition, Figures 7g to i are not discussed in the main text.

Responses: Thanks for your reminder. The constant cell voltage value is 1.56 V versus RHE. This has been corrected in the revised paper. Figures 7g to 7i have been discussed in the revised paper.

Reviewer #2:

The study of green energy has become an increasingly important issue due to the change of climate and the exhaustion of fossil fuels. Splitting of water into hydrogen and oxygen is a

sustainable and very important technology for energy conversion and storage. Oxygen and hydrogen evolution reactions are the basic half reaction for water splitting. Developing efficiency, inexpensive and sustainable catalyst plays a key role for water splitting reaction. In this work, the authors developed a comprising graphdiyne and iron/cobalt layered double-hydroxide nanosheet arrays and applied as the oxygen and hydrogen evolution reactions and even a bifunctional electrocatalyst. The catalyst presents low overpotential and long-term stability and higher catalytic activity than other reactive catalyst. The manuscript is well organized and solid. I believe this important study will provide a new idea for GDY application. Therefore, I recommend it to publish in *Nature Communication* with the following revisions:

1. The authors state that they combine graphdiyne (GDY) with exfoliated iron/cobalt LDH nanosheet arrays grown on nickel foam. The LDH was exfoliated through the intercalating of monomer (HEB). How does it work?

Responses: Layered double hydroxides (LDHs) are a class of ionic lamellar compounds made up of positively charged brucite-like layers with an interlayer region containing charge compensating anions and solvation molecules. In general, the anions located in the interlayer regions can be easily replaced. And the large interlayer distance in LDH make it easy for the entrance of hexaethynylbenzene (HEB) monomers into the LDHs gallery. The acetylenic hydrogen of HEB could then form hydrogen bonds with the hydroxide layer, which leads to the formation of HEB film on the LDH surfaces. Copper is easy to convert to Cu ions in the presence of a catalytic amount of base, where a Glaser–Hay reaction took place efficiently with the aid of TMEDA (*J. Am. Chem. Soc.* **2015**, *137*, 7596–7599; *Adv. Mater.* **2016**, *28*, 168–173; *Adv. Mater.* **2017**, *29*, 1605308). The GDY films could be uniformly grown on LDH surfaces. Owing to the flexibility nature of both GDY and LDH, the stress/deformation caused by the intimate contact in-between will further enlarge the layer spacing, leading to the complete exfoliation of b-LDHs into thicker LDH nanosheets (e-LDH). The resulted product is denoted here as e-LDH@GDY/NF.

Corresponding discussions have been added in the revised manuscript.

Page 6 lines 1-9:

In general, the anions located in the interlayer regions can be easily replaced. And the large

interlayer distance in LDH make it easy for the entrance of hexaethynylbenzene (HEB) monomers into the LDHs gallery. The acetylenic hydrogen of HEB could then form hydrogen bonds with the hydroxide layer, which can lead to the formation of HEB film on the LDH surfaces. A Glaser–Hay reaction would then take place efficiently with the aid of TMEDA. The GDY films could be uniformly grown on LDH surfaces. Owing to the flexibility nature of both GDY and LDH, the stress/deformation caused by the intimate contact in-between will further enlarge the layer spacing, leading to the complete exfoliation of b-LDHs into thicker LDH nanosheets (e-LDH). The resulted product is denoted here as e-LDH@GDY/NF.

2. According to the schematic representation (Fig. 1), the LDH nanosheets were exfoliated into several thicker nanosheets (from 80 nm to 7 nm). Actually, from SEM images in Fig. 2, it is the same that the number of nanosheet arrays of LDH is not increase after GDY growth. The authors should give the reasons. Is it possible other reasons? Such as the corrosion of pyridine or TMEDA, which makes the LDH layer more thicker.

Responses: As can be seen from Fig. 2 in the manuscript, it is clear that the number of e-ICLDH@GDY nanosheets is much larger than that of pristine ICLDH nanosheets. The density of e-ICLDH@GDY nanosheets was roughly calculated to be ~ 70 nanosheet per μm^2 , which is much larger than that of pristine ICLDH nanosheets (~ 18 nanosheet per μm^2). In addition, pristine ICLDH nanosheets exhibit a thickness of approximately 40–80 nm and a smooth surface. After the cross-coupling reaction, nanosheets with thinner thickness (around 7.0 nm) and more wrinkled morphology were observed.

In order to determine whether the pyridine or TMEDA can damage the morphology of the LDHs, pristine LDHs were put into the reactor containing pyridine or TMEDA only (without HEB). Other experimental conditions were the same as that used for synthesizing e-LDH@GDY/NF. SEM images exhibited almost no changes in morphology before and after the treatments (Fig. R4 and R5). These results further indicated the GDY-induced exfoliation of the bulk LDHs. Based on these facts, we employed the schematic diagram for illustrating the e-LDH@GDY/NF synthetic process. These discussions have also added in the revised manuscript as presented here.

In order to confirm whether the morphologies of the LDHs in pyridine and TMEDA can be damaged, pristine LDHs were put into the reactor containing pyridine or TMEDA only (without HEB). Other experimental conditions were the same as that used for synthesizing e-ICLDH@GDY/NF. SEM images exhibited no any changes in morphology before and after the treatments by pyridine or TMEDA (Supplementary Figs. 1 and 2).

Figure R4. SEM images of ICLDH/NF (a, b) before and (c, d) after being treated by pyridine.

Figure R5. SEM images of ICLDH/NF (**a, b**) before and (**c, d**) after being treated by TMEDA.

3. What are the thickness of GDY film and LDH?

Responses: Figure R6 shows HRTEM image of the lateral standing e-ICLDH@GDY nanosheet. Because of the imaging angle, the GDY layer coated on the other side was not observed. The thickness of GDY (or LDH) is estimated from the number of fringes corresponding to the distance between two consecutive carbon (or LDH) fringes. Therefore, the thickness of GDY and LDH are determined to be about 1.2 nm and 5.1~6.5 nm, respectively. Corresponding contents were also added in the revised text as follows.

Page 10 lines 11-14:

Supplementary Fig. 3 shows the HRTEM image of the lateral standing e-ICLDH@GDY

nanosheet. The thickness of GDY and LDH are determined to be about 1.2 nm and 5.1-6.5 nm, respectively, which is in accordance with the SEM (Fig. 2i) and AFM imaging (Figs. 2o and 2p) results.

Figure R6. HRTEM image of the lateral standing e-ICLDH@GDY nanosheet. The thickness of GDY and LDH are determined to be about 1.2 nm and 5.1~6.5 nm, respectively.

4. In the FT-IR spectra, a sharp peak presents in 2103 cm^{-1} . It corresponds to the stretching of $\text{C}\equiv\text{C}$ bonds. According to your previous reported work, the peak at 2103 cm^{-1} is agree with the terminal alkynes and IR signal of the $\text{C}\equiv\text{C}$ bonds in GDY should be as weak as possible because of the molecular perfectly symmetry. How to explain the sharp peak presents in 2103 cm^{-1} ?

Responses: The peak at 2103 cm^{-1} corresponds to the internal alkyne stretch. Our previous reported work (*Chem. Commun.*, 2010, 46, 3256–3258) also indicated that the band appeared here corresponded to the typical $\text{C}\equiv\text{C}$ stretching (not the terminal alkynes).

In fact, the signal indicative of the presence of $\text{C}\equiv\text{C}$ in the FT-IR spectra is really very weak. As shown in Fig. 4a (the original spectra of our samples), it is very hard to see signals in the region of 2050 to 2150 cm^{-1} . In order to make the readers more clearly see, we magnified this region as shown in Fig. 4c. But it is sorry that the over-magnification leads to a misunderstanding. The updated figure with proper magnification was shown in Fig. R7c.

Figure R7. (a) FTIR spectra of the e-ICLDH@GDY/NF (red line) and pristine ICLDH/NF (blue line) structures. (b) Enlarged image of the selected area of the FTIR spectrum in a. (c) Selective enlargement of the FTIR spectra of e-ICLDH@GDY/NF. (d) XPS survey spectrum of e-ICLDH@GDY/NF. (e, f) Core-level XPS spectra of the (e) C 1s and (f) O 1s binding energies of the e-ICLDH@GDY/NF and pristine ICLDH/NF electrocatalysts.

5. According to the electrochemical testing, whatever OER, HER or Overall water splitting, the catalytic activity of e-ICLDH@GDY/NF is better than those of previously reported LDH-related catalysts and nearly close to the Pt. Does it is possible to be used in practical applications? What are the challenges?

Responses: It is very good question. The material of e-ICLDH@GDY/NF is easy to prepare, easy to treatment, and shows excellent catalytic activity, which is one of the most promising electrocatalysts for practical applications. The combined material not only shows excellent catalytic activities but also exhibit robust, long-term stability. These superior properties are rarely found together, and their combined effect make it very possible to be used in practical applications.

According to our results and previously reported literatures, the combination of monolayer LDHs and monolayer GDY should have a great potential to further enhance the catalytic activity. This is still a big challenge. Related research work is now underway.

6. The authors used two classic synthetic methods (GDY nanowall synthesis and GDY synthesis on arbitrary substrates) for GDY preparation. The related literatures should be cited (J. Am. Chem. Soc. 2015, 137, 7596; Adv. Mater. 2017, 29, 1605308).

Responses: The related literature have been cited in the revised text.

7. There are several mistakes about Pt or RuO₂ in Fig. 8 and 9 of SI.

Responses: Corrections have been made in the revised manuscript.

Reviewer #3:

The manuscript “Overall water splitting by graphdiyne-exfoliated and -sandwiched layered double-hydroxide nanosheet arrays” demonstrates a highly active catalyst composed of graphdiyne and FeCo LDH for OER and overall water electrolysis. The performance of the electrodes is much higher than that of majority of previous investigations. Here, some points are still not clear enough to understand this work and need further explanation is necessary to help comprehend it.

1. In the experimental section, Cu foils were added during the preparation of e-ICLDH@GDY. What is the effect of Cu foils? Is it possible to dope Cu into e-ICLDH@GDY during this process? More, in supplementary figure 2, the peak at ~8.9 keV in EDS spectrum is not indexed to Co, Ni or Fe. However, it is very close to Cu.

Responses:

- In the experiment, Cu foils can easy release Cu ions in the presence of a catalytic amount of base to form Cu-based catalysts for the Glaser–Hay reaction.
- Cu species cannot be doped into the e-ICLDH@GDY structure. In order to avoid the copper contamination, all e-ICLDH@GDY/NF products were washed with 2 M HCl, 2 M NaOH, acetone, DMF and deionized water subsequently before use (please see Methods section in the manuscript for details). Moreover, XPS (Fig. 4), HRTEM (Fig. 3) and EDS (Supplementary Figure 4) results also demonstrated that there is no Cu species present in the e-ICLDH@GDY nanosheets.

- It needs to be mentioned that the EDS spectrum shown in Supplementary Figure 2 (Supplementary Figure 4 in the revised manuscript) was measured on a molybdenum (Mo) grid. We have discussed with the engineer regarding the peak located at ~8.9 keV. She said that this peak was ascribed from to the TEM system, for example, the sample rod. In order to clarify this issue, a control measurement on pure molybdenum grid was conducted. The obtained EDS result is shown in Fig. R8.

Figure R8. (a) STEM image and (b) corresponding EDS pattern of the pure molybdenum grid. Inset of b: the relative content of each element.

2. Considering the reduced thickness of nanosheets from tens of nanometers to several nanometers. The volume expansion should be very large. However, this wasn't observed based on SEM images in Figure 2. Then, is there any loss of the electroactive materials during the growth of GDY?

Responses: The volume expansion phenomenon was observed by using STEM measurement, as shown in Fig. R9. With the purpose of examining whether there is any loss of LDH during the reaction process, ICP measurements for pristine ICLDH/NF and ICLDH@GDY/NF have been conducted, showing the reduction of the Fe and Co contents. In addition, XPS measurements was conducted on the residues obtained after the synthesis of electrocatalysts (Fig. R10). Signals corresponds to Fe and Co elements were observed. These results indicate the peeling off of LDH species during the GDY growing process, which could be due to the volume expansion effect due to the GDY intercalation and exfoliation.

Figure R9. (a) Typical scanning TEM and corresponding elemental mapping images of (b) overlapping, (c) C, (d) O, (e) Fe, and (f) Co atoms in the e-ICLDH@GDY nanosheets.

Figure R10. (a) Fe 2p and (b) Co 2p core-level XPS spectra of the residues obtained after the synthesis of electrocatalysts.

3. The e-ICLDH@GDY/NF catalyst exhibited good stability for OER, and how is the chemical state of GDY after OER stability test? Is there any possibility of being oxidized for GDY?

Responses: The chemical state of the sample after OER durability tests has been examined by using XPS measurements (Fig. R11). In the C 1s spectrum, compared with pristine GDY, there is a slightly increasing in the relative content of O=C-OH from 9.9 % to 15.3 % (located at

288.6 eV), indicating the slight oxidization of GDY after stability test.

Figure R11. Core-level XPS spectra of the C 1s of the e-ICLDH@GDY/NF obtained after OER stability test.

4. What are the difference and relationship between figure 6g and supplementary figure 8, which is very confusing? In figure 6, the e-ICLDH@GDY/NF sample performed best for HER, but in figure s8, the activity of e-ICLDH@GDY/NF was very poor and ICLDH/NF performed very well for HER. And in figure 6g/6h and figure s9, the labels of Pt/RuO₂ are very confused. Please be careful on those labels, which can significantly mislead the readers.

Responses: Thanks for your advice. There are mistakes in the labels of samples in Figure 6h, Supplementary Figure 8 and Supplementary Figure 9. These have been corrected in the revised manuscript and Supplementary materials.

5. In figure 6k and figure s11, the fitting data are not clear enough with circles, lines may be better and preferred.

Responses: These have been changed in the revised text.

6. Are the CV curves in figure 7b iR-corrected or not? If they are iR-corrected, it is better to

supply the pristine data without *iR* corrections. For overall water splitting, polarization curves without *iR* corrections are more convincing from the view of practical applications. The cell voltage of 0.5V should be double confirmed in figure 7i, which is unbelievable to drive such a high current density for overall water splitting.

Responses: The CV curves in Figure 7b were *iR*-corrected. According to the review's suggestion, polarization curves without *iR* corrections are provided (Fig. R12).

The cell voltage in Figure 7i is 1.56 V versus RHE. This has been corrected in the revised text.

These results have been updated in the revised manuscript.

Figure R12. CV curves (without *iR*-correction) for overall water splitting.

7. The authors claimed that "...a greater density of electrons at the electrocatalyst surface should be of more help for the formation of *OOH and thus enhance the OER activity..." (line 230 and 231). However, in figure 5g and 5h, results showed that both Co and Fe suffered great electron transfer to GDY, which is also the result of theoretical calculations. It seems a paradox between the results and analysis. How to understand the effect of electron transfers between metal and GDY on the catalytic activity? More discussions may be given here to clarify this effect for both OER and HER.

Responses: It seems that there was a misunderstanding about the chemical structure of our reported electrocatalyst (e-ICLDH@GDY). As detailed described in the manuscript, the synthesized e-ICLDH@GDY comprises of ICLDH core and GDY shell. Our experimental and theoretical results showed that the great electron transfer from Fe and Co to GDY could result

in the increase of the electron density in GDY structure (the surface structure of our electrocatalyst). For OER, e-ICLDH@GDY has a smaller free energy of 0.54 eV for the formation of *OOH species than pure GDY (0.93 eV, Fig. R13), which indicates an enhanced OER activity for e-ICLDH@GDY. For HER, e-ICLDH@GDY possesses the smallest Gibbs free energy (0.34 eV) compared to pure GDY (0.46 eV) and ICLDH (1.55 eV), indicating the best HER activity for GDY modified ICLDH (Fig. R14).

Corresponding discussions have been added in the revised manuscript as follows:

Page 15 lines 3-5:

the obvious electron transfer from LDH to GDY in e-ICLDH@GDY should, therefore, favor the formation of *OOH species with a smaller free energy of 0.54 eV than that of pure GDY (0.93 eV, Fig. 5i) and benefit the OER activity.

Page 15 lines 8-10:

Our calculated results (Fig. 5j) further show that the ICLDH@GDY has the smallest Gibbs free energy of 0.34 eV for HER compared with that of pristine GDY (0.46 eV) and ICLDH (1.55 eV), indicating the excellent HER activity for GDY incorporated ICLDH.

Figure R13. The free energy changes for the formation of OOH* and corresponding stable structures of GDY (ΔG_1) and e-ICLDH@GDY (ΔG_2).

Figure R14. Gibbs free energy profiles of HER and corresponding stable structures of e-ICLDH@GDY, GDY and ICLDH, respectively.

8. In this work, the very large catalytic current densities for both HER and OER are driven by small potentials, which is very impressive. In this case, the Faraday efficiencies of OER, HER and overall water splitting should be given to further confirm the performance of electrodes.

Responses: The faradaic efficiency was calculated by comparing the experimentally observed gas amount (H_2 for HER; O_2 for OER and OWS) with that obtained by the theoretical results (Fig. R15). The faradaic efficiency for HER, OER and OWS reached nearly $98.96 \pm 0.86\%$, $97.98 \pm 0.81\%$ and $97.40 \pm 1.30\%$, respectively.

Figure R15. Current-time curves and Faradaic efficiencies of e-ICLDH@GDY/NF for (a,b) HER, (c,d) OER and (e,f) OWS, respectively.

Thank you again for your kind consideration.

Best wishes,

Yuliang Li

Reviewers' comments:

Reviewer #1 (Remarks to the Author):

The authors addressed some of my concerns, and the manuscript is much improved. However, some questions are still not clearly illustrated. A major revision is still required.

1. HER in alkaline media is more complex and sluggish than that in acid due to an additional water activation step, which is normally the rate-limiting step. Therefore, it is not accurate to evaluate the HER activity just by ΔG . The authors can refer to the following paper and carry out similar computations (Nature Communications (2017) 8, 15437; Nano Energy (2018) 53, 492).
2. For the TOF calculations, the authors took the molar number of metal ions on the electrode as N, so the calculated TOF of HER and OER was very large. This is not solid because the metal ions are not the only active sites. The authors should reconsider this issue and provide calculation details in the revised manuscript.
3. The authors' response to my sixth question in the previous review comments is still not clear. Why will the surface complexity lower the cell voltage for two-electrode water splitting?
4. If the oxidation peak corresponds to the oxidation of Co^{2+} to Co^{3+} , why it is absent for the other samples (ICLDH, CLDH@GDY, and CLDH)? In fact, the oxidation peak of $\text{Co}^{2+}/\text{Co}^{3+}$ should be located at 1.3 vs. RHE (Advanced Energy Materials (2018), 8, 1701694).
5. The polarization curves in Figures 7b and 7c are very different, and Figure 7c seems over-compensated. Please provide more details on compensation during LSV tests.

Reviewer #3 (Remarks to the Author):

I have went through the revised manuscript. Authors have well responded all my comments. I have not more questions. It could be accepted as the current version.

Dear Reviewers,

Thank you for the sincere advice and comments on our manuscript titled “Overall water splitting by graphdiyne-exfoliated and -sandwiched layered double-hydroxide nanosheet arrays” for *Nature Communications*. Based on your comments, we have made the corresponding revisions on the manuscript and the following point-by-point responses to the comments.

Reviewer #1:

The authors addressed some of my concerns, and the manuscript is much improved. However, some questions are still not clearly illustrated. A major revision is still required.

Response: Thank you for your time and detailed comments on our manuscript. We have carefully addressed all your comments and make necessary revisions on our manuscript to address your concerns. The point-by-point responses are as follows.

Question 1. HER in alkaline media is more complex and sluggish than that in acid due to an additional water activation step, which is normally the rate-limiting step. Therefore, it is not accurate to evaluate the HER activity just by ΔG . The authors can refer to the following paper and carry out similar computations (Nature Communications (2017) 8, 15437; Nano Energy (2018) 53, 492).

Response: We thank the reviewer for the valuable comments. Here we further gained some insightful interpretations on the high alkaline HER performance on the HER by DFT calculations (**Figure R1**). The projected partial density of states (PDOSs) show the related p- and d- band distributions for the nearest interfacing GDY and LDH layers. The overall 3d-band center of ICLDH is high next to the Fermi level (E_F), which is responsible for activating the initial adsorptions of O-species from alkaline condition. The partial intrinsic overlaps of p- and d- bands demonstrate a weak inter-layer bonding. This indicates a rather long-ranged interacting interfacing layer with large distance for H₂O molecule easily passing through, without obviously impacting the catalyst interface (**Figure R1a**).

Details on the 3d-bands of Co and Fe sites from LDH interfacing layer show the contrast in 3d-band centers, which potentially denotes a dynamically self-activated electronic activity of redox. The Co-3d bands with majority-spin is pinned at the E_F and locates 1.4 eV higher than one of Fe-3d band. This trends show Fe-3d acts as electron-rich center while the Co-3d band plays as electron-depletion channel. However, their d-d overlapping is rather weak with few overlaps in PDOS. This indicates the highly effective electron-transfer occurs between the inter-layers instead of intra-layer ionic sites. The O-2p band from H₂O exhibits a large overlapping across

the Fe-3d band confirming the Fe-site acts as dominant role for H₂O initial adsorption within interlayer regions (**Figure R1b**).

We move onto the energetic pathway of the alkaline HER for this interface system. Overall, the e-ICLDH@GDY system performs the most energetic favorable path with lowest adsorption (-0.56 eV) and preferred exothermic reaction heat for H₂ (-1.40 eV). In the contrast, the GDY exhibits the highest barrier for initial H₂O adsorption (1.92 eV) and an endothermic reaction (0.23 eV) for HER. The capabilities of water-splitting demonstrate the preference as e-ICLDH@GDY > ICLDH > GDY, respectively (**Figure R1c**).

We also compared the intermediating H₂O-splitting and the transition state barriers are 0.28 eV (e-ICLDH@GDY), 0.44 eV (ICLDH), and 0.65 eV (GDY), respectively (**Figure R1d**). Meanwhile, the chemisorption energy of HER has been also illustrated. Among of them, the e-ICLDH@GDY indicates even more energetic favorable trend for HER compared with pristine-GDY, while the LDH meets the unflavored uphill for H-adsorption (**Figure R1e**). From the local structures (**Figure R1f**), the stable adsorbed H is bonding with C2 site at the GDY layer and the H₂O location will pass through the GDY locating near Fe-site on the ICLDH surface. The splitting H₂O process occurs between the C2 site and Fe-site from LDH layer. The two closely adsorbed H exhibit a favorable trend for combination in structural configuration for potentially efficient H₂ generation. With assistance of C2 site, the leaving group (OH⁻) can be stably located on the surface of GDY.

We have added the above results and the suggested References in the revised manuscript (page 15 lines 13-25 and page 16 lines 1-20).

Figure R1. (a) PDOSs of the 3d and 2p bands of interfaced system containing GDY and ICLDH layers. (b) PDOSs of Fe-3d, Co-3d, H₂O-s and H₂O-p bands near the interface region. (c) Energetic pathway of HER under alkaline conditions for e-ICLDH@GDY, ICLDH, and GDY, respectively. (d) Comparison on the transition state barrier for H₂O-splitting among three systems. (e) H-chemisorption of these three systems. (f) Structural evolution path for alkaline HER within e-ICLDH@GDY interface system.

2. For the TOF calculations, the authors took the molar number of metal ions on the electrode as N, so the calculated TOF of HER and OER was very large. This is not solid because the metal ions are not the only active sites. The authors should reconsider this issue and provide calculation details in the revised manuscript.

Response: According to the previously reported method (*Nat. Commun.* 2017, 8, 15437; *Nano Energy* 2018, 53, 492; *Nat. Commun.* 2016, 7, 12765; *Angew. Chem. Int. Ed.* 2014, 53, 14433), we carried out similar calculation method. The number of active sites was estimated from the roughness factor (eq. 1).

$$\frac{\# \text{ Surface sites}}{\text{cm}^2 \text{ geometric area}} = \frac{\# \text{ Surface sites (flat standard)}}{\text{cm}^2 \text{ geometric area}} \times \text{Roughness factor} \quad (1)$$

The roughness factor (R_f) can be determined by the electrochemically double-layer capacitance (C_{dl}). The specific capacitance can be converted into an electrochemical active surface area (ECSA) using the specific capacitance value for a flat standard with 1 cm² of real surface area. According to previous reports (*Energy Environ. Sci.* 2015, 8, 3022; *Nano Energy* 2018, 53, 492), we assume 60 μF cm⁻² for a flat electrode and the surface sites of 2 × 10¹⁵ for the flat standard electrode. As a result, the number of surface active sites for e-ICLDH@GDY is calculated to be 0.067 × 10¹⁸ surface sites/cm².

The TOF values can then be obtained according to the following formulas:

$$\text{TOF} = \frac{\# \text{ Total Hydrogen Turn Overs/cm}^2 \text{ geometric area}}{\# \text{ Surface Sites /cm}^2 \text{ geometric area}} \quad (2)$$

The number of total hydrogen turn overs is calculated from the current density extracted from the LSV curves:

Total Hydrogen Turn Overs

$$\begin{aligned} &= \left(j \frac{\text{mA}}{\text{cm}^2} \right) \left(\frac{1 \text{C s}^{-1}}{1000 \text{mA}} \right) \left(\frac{1 \text{mol e}^-}{96485.3 \text{C}} \right) \left(\frac{1 \text{mol H}_2}{2 \text{mol e}^-} \right) \left(\frac{6.022 \times 10^{23} \text{H}_2 \text{ molecules}}{1 \text{mol H}_2} \right) \\ &= 3.12 \times 10^{15} \frac{\text{H}_2/\text{s}}{\text{cm}^2} \text{ per } \frac{\text{mA}}{\text{cm}^2} \end{aligned} \quad (3)$$

TOF per site for e-ICLDH@GDY at different overpotentials (η) is calculated as follows:

$$\text{TOF} = \frac{3.12 \times 10^{15}}{0.067 \times 10^{18}} \times j = 0.047 \times j \quad (4)$$

j corresponds to the current density at different overpotentials.

The C_{dl} of pure ICLDH is 0.6 mF cm⁻². According to Eq. (1), the number of surface active sites for pure ICLDH is calculated to be 0.02 × 10¹⁸ surface sites/cm². Thus, TOF per site for e-ICLDH@GDY at different overpotentials (η) can be calculated as follows:

$$\text{TOF} = \frac{3.12 \times 10^{15}}{0.02 \times 10^{18}} \times j = 0.156 \times j \quad (4)$$

For HER, at the overpotentials of 50, 100, and 200 mV, the j for e-ICLDH@GDY are 11.01, 20.69, and 179.16 mA cm⁻² respectively. For pure ICLDH, The j at $\eta = 50, 100, \text{ and } 200 \text{ mV}$ are 1.40, 1.58, and 3.55 mA cm⁻², respectively. As summarized below, the e-ICLDH@GDY

exhibits much bigger TOF value than the pure ICLDH, suggesting its higher HER catalytic activity.

Catalysts		$\eta = 50 \text{ mV}$	$\eta = 100 \text{ mV}$	$\eta = 200 \text{ mV}$
HER	e-ICLDH@GDY	0.517 s^{-1}	0.944 s^{-1}	8.44 s^{-1}
	ICLDH	0.218 s^{-1}	0.246 s^{-1}	0.554 s^{-1}

By using the same calculation method, the TOF values of OER was obtained. At $\eta = 250 \text{ mV}$, the TOF values for e-ICLDH@GDY and pure ICLDH are 2.34 and 1.29 s^{-1} , respectively. This result indicates the better OER catalytic activity of e-ICLDH@GDY than pure ICLDH.

The calculation details were added in the Supplementary Methods section. Corresponding discussions have also been added in the revised text as below.

Page 23 lines 3-7:

Turnover frequencies (TOFs) have also been used to assess the information about the intrinsic activity (see Supplementary Methods for details). The e-ICLDH@GDY/NF shows TOF values 8.44 s^{-1} at 200 mV for HER and 2.34 s^{-1} at 250 mV for OER, respectively, which are larger than that of pure ICLDH and most of recently reported catalysts (Supplementary Table 5).

3. The authors' response to my sixth question in the previous review comments is still not clear. Why will the surface complexity lower the cell voltage for two-electrode water splitting?

Response: It is a very good question.

For overall water splitting (OWS), the two-electrode cell system was employed to conduct corresponding measurement. In the two-electrode cell system, two electrodes are the same e-ICLDH@GDY/NF samples, one is used as the working electrode and the other is used as counter/reference electrode, to measure the potential across the complete cell. It should be mentioned that, compared to the SCE electrode (a standard electrode), the prepared e-ICLDH@GDY/NF electrodes have different sizes, shapes, and morphologies. The distances between the WE and the RE are different, which would result in different solution resistances. Accordingly, the cell voltages in OWS system is different from that for OER and HER.

This phenomenon can also be found in some literatures, including *Nat. Commun.* **9**, 2452 (2018); *Nat. Commun.* **8**, 15377 (2017); *Nat. Commun.* **6**, 7261 (2015); *J. Am. Chem. Soc.* **140**, 2610–2618 (2018); *Angew. Chem. Int. Ed.* **56**, 573–577 (2017); *Adv. Mater.* **29**, 1606200 (2017); *Adv. Mater.* **29**, 1701584 (2017); *Adv. Mater.* **29**, 1700017 (2017); *Energy Environ. Sci.* **10**, 1820–1827 (2017).

4. If the oxidation peak corresponds to the oxidation of Co^{2+} to Co^{3+} , why it is absent for the other samples (ICLDH, CLDH@GDY, and CLDH)? In fact, the oxidation peak of $\text{Co}^{2+}/\text{Co}^{3+}$ should be located at 1.3 vs. RHE (*Advanced Energy Materials* (2018), **8**, 1701694).

Response: As shown in **Figure R2**, after the magnifications the oxidation peaks corresponding to the oxidation of Co^{2+} to Co^{3+} for ICLDH/NF, CLDH@GDY/NF, and CLDH/NF samples are clearly observed.

The CV for pure CLDH/NF and CLDH@GDY/NF exhibit two pronounced peaks in the anodic process (**Figure R2b**). According to some previous literatures (*Phys. Chem. Chem. Phys.* 2015, **17**, 29387; *Chem. Commun.* 2015, **51**, 7851; *ACS Nano* 2012, **6**, 3206; *Electrochim. Acta* 2000, **45**, 4359), the first peak located at around 1.28 V mainly result from the oxidation of Co^{2+} to Co^{3+} , and the latter at 1.36 V corresponds to oxidation of Co^{3+} to Co^{4+} . In addition, for pure ILDH (**Figure R2c**), the oxidation peak was observed at round 1.41 V (*Adv. Mater.* 2015, **27**, 3305.). The CV of e-ICLDH@GDY/NF (**Figure R2d**) shows only one broad anodic peak at around 1.25 V and a cathodic peak at around 1.16 V. The redox behavior of the e-ICLDH@GDY/NF sample is mainly attributed to the $\text{Co}^{2+}/\text{Co}^{3+}$ and $\text{Co}^{3+}/\text{Co}^{4+}$ redox pairs and contribution from iron (*J. Am. Chem. Soc.* 2011, **133**, 5587; *J. Appl. Electrochem.* 2009, **39**, 2469; *J. Phys. Chem. C* 2011, **115**, 15646; *J. Mater. Chem. A* 2015, **3**, 16849.). This has already been reported in alkaline solutions. Such high charge densities would be particularly beneficial for the improvement of the OER catalytic activity.

Corresponding discussions have been added and highlighted in the revised text as below.

Page 17 lines 14-22:

As shown in Supplementary Figure 12, the CVs for pure CLDH/NF and CLDH@GDY/NF exhibit two pronounced oxidation peaks at 1.28 and 1.36 V corresponding to the oxidation of

Co²⁺ to Co³⁺ and the oxidation of Co³⁺ to Co⁴⁺, respectively^{42–44}. For pure ILDH, the oxidation peak was observed at round 1.41 V⁴⁵. The CV of e-ICLDH@GDY/NF shows only one broad anodic peak at around 1.25 V and a cathodic peak at around 1.16 V (Supplementary Fig. 11d). The redox behavior of the e-ICLDH@GDY/NF sample is mainly attributed to the Co²⁺/Co³⁺ and Co³⁺/Co⁴⁺ redox pairs and contribution from iron^{43,46–48}. Such high charge densities would be particularly beneficial for the improvement of the OER catalytic activity^{43,46}.

Figure R2. (a) OER cyclic voltammetry (CV) curves for e-ICLDH@GDY/NF, ICLDH/NF, ILDH@GDY/NF, ILDH/NF, CLDH@GDY/NF, CLDH/NF, and GDY/NF, respectively. (b) Magnifications of the pure CLDH/NF and CLDH@GDY/NF curves. (c) Magnifications of the pure CLDH/NF and ILDH/NF curves. (d) Magnifications of the ICLDH/NF and e-ICLDH@GDY/NF curves.

5. The polarization curves in Figures 7b and 7c are very different, and Figure 7c seems over-compensated. Please provide more details on compensation during LSV tests.

Response: It seems that there was a misunderstanding about the polarization curves Figures 7b and 7c. All polarization curves illustrated in Figure 7b are the pristine data without iR corrections. But the polarization curves shown in Figure 7c are iR -corrected.

The polarization curves were iR -compensated according to the equation ($E_{\text{calibration}} = E_{\text{original}} - iR$, where i is the current flowing and R is the solution resistance measured from the EIS Nyquist plot).

Reviewer #3 (Remarks to the Author):

I have went through the revised manuscript. Authors have well responded all my comments. I have not more questions. It could be accepted as the current version.

Response: Thank you!

Thank you again for your positive comments on our manuscript. Should there been any other corrections we could make, please feel free to contact us.

Yours Sincerely,

Yuliang Li

REVIEWERS' COMMENTS:

Reviewer #1 (Remarks to the Author):

The manuscript has been greatly improved and can be accepted for publication now.